

# Assimilation of Transformed Water Surface Elevation to Improve River Discharge Estimation in a Continental-Scale River

Menaka Revel[1], Xudong Zhou[1], Dai Yamazaki[1], Shinjiro Kanae[2]

[1]Global Hydrological Prediction Center, Institute of Industrial Science, The University of Tokyo, Tokyo, 153-8505, Japan
[2]Department of Civil and Environmental Engineering, Tokyo Institute of Technology, Tokyo, 152-8550, Japan

*Correspondence to*: Menaka Revel (menaka@rainbow.iis.u-tokyo.ac.jp)

**Abstract.** Quantifying continental-scale river discharge is essential to understanding the terrestrial water cycle but is susceptible to errors caused by a lack of observations and the limitations of hydrodynamic modeling. Data assimilation (DA) methods are increasingly used to estimate river discharge in combination with emerging river-related remote sensing products
(e.g., water surface elevation [WSE], water surface slope, river width, and flood extent). However, directly comparing simulated WSE to satellite altimetry data remains challenging (e.g., because of large biases between simulations and observations or uncertainties in parameters), and large errors can be introduced when satellite observations are assimilated into hydrodynamic models. In this study we performed direct, anomaly, and normalized value assimilation experiments to investigate the capacity of DA to improve river discharge within the current limitations of hydrodynamic modeling. We
performed hydrological DA using a physically-based empirical localization method applied to the Amazon Basin. We used satellite altimetry data from ENVISAT, Jason 1, and Jason 2. Direct DA was the baseline assimilation method and was subject to errors due to biases in the simulated WSE. To overcome these errors, we used anomaly DA as an alternative to direct DA. We found that the modeled and observed WSE distributions differed considerably (e.g., differences in amplitude, seasonal flow variation, and a skewed distribution due to limitations of the hydrodynamic models). Therefore, normalized value DA
was performed to improve discharge estimation. River discharge estimates were improved at 24%, 38%, and 62% of stream gauges in the direct, anomaly, and normalized value assimilations relative to simulations without DA. Normalized value assimilation performed best for estimating river discharge given the current limitations of hydrodynamic models. Most gauges within the river reaches covered by satellite observations accurately estimated river discharge, with Nash-Sutcliffe efficiency ($NSE$) > 0.6. The amplitudes of WSE variation were improved in the normalized DA experiment. Furthermore, in the
Amazon Basin, normalized assimilation (median $NSE = 0.50$) improved river discharge estimation compared to open-loop simulation with the global hydrodynamic model (median $NSE = 0.42$). River discharge estimation using direct DA methods was improved by 7% with calibration of river bathymetry based on $NSE$. The direct DA approach outperformed the other DA approaches when runoff was considerably biased, but anomaly DA performed best when the river bathymetry was erroneous. The uncertainties in hydrodynamic modeling (e.g., river bottom elevation, river width, simplified floodplain dynamics, and
the rectangular cross-section assumption) should be improved to fully realize the advantages of river discharge DA through the assimilation of satellite altimetry. This study contributes to the development of a global river discharge reanalysis product that is consistent spatially and temporally.

## 1   Introduction

River discharge plays a pivotal role in the  global water cycle and thereby affects human livelihoods (Oki and Kanae, 2006).
River discharge can be used to assess water resources, biogeochemistry, and the carbon cycle in terrestrial waters and is the single most important parameter affecting the flow dynamics of rivers (Gleason and Durand, 2020). The ability to measure global river discharge via insitu gauging is limited by a lack of accurate, complete, and freely available data (Hannah et al., 2011; Shiklomanov et al., 2002; Vörösmarty et al., 2001). Because of the limited temporal coverage and spatial heterogeneity of insitu gauging networks, elucidating the terrestrial water cycle is essential.



As a result of recent computational advances, global hydrologic/hydrodynamic models (GHMs) have been used extensively to study the terrestrial water cycle (Döll et al., 2016; Sood and Smakhtin, 2015). Simulated water dynamics obtained from GHMs are used to compensate for unavailable ground observations. GHMs simulate water dynamics in discretized river segments to increase computational efficiency. Nevertheless, they are subject to numerous limitations, including simplified model structures, imperfect external forcing, and uncertainties in model parameters (Liu and Gupta, 2007; Renard et al., 2010).

These inadequacies are due to both a lack of information about physical processes and simplifications made to limit computational costs. There are considerable uncertainties in model parameters such as river bottom elevation due to a lack of measurements or limitations of estimation methods that affect model outputs (Brêda et al., 2019). Uncertainties in the forcing factors (i.e., runoff) are also partially responsible for uncertainty in the surface water dynamics (Emery et al., 2020c). In combination, these constraints result in unavoidable uncertainties in GHM simulations of water dynamics.

Given the current limitations of GHMs, satellite altimetry observations provide an alternative method of estimating surface water dynamics (Feng et al., 2021). Satellite altimetry quantifies the water surface elevation (WSE) by measuring the time required for the radar pulse to travel between the satellite and the water surface. Beginning with GEOS-3 in 1975, numerous satellite altimetry missions have been deployed to obtain measurements of terrestrial water surfaces. Although some of these satellites were developed for other purposes (i.e., observing the sea surface), their application has expanded to include river

and lake observations (Birkett et al., 2002; Crétaux et al., 2009; Santos da Silva et al., 2010). Commonly used satellite missions for river observations are ENVISAT, Jason 1, Jason 2, Sentinel 3A, and Sentinel 3B (Bannoura, 2001; Resti et al., 2002; Zwally et al., 2002). The Surface Water and Ocean Topography (SWOT) satellite will provide an unprecedented amount of data on surface waters (Biancamaria et al., 2016; Fu et al., 2012). The greatest impediment to the use of these satellites is their limited spatial and temporal coverage, which ranges from a few days to several months between successive observations of

specific locations. Hence, satellite altimetry observations may not provide a comprehensive view of the terrestrial water cycle because of their spatial and temporal sparseness.

Surface water dynamics can be clarified by combing remote sensing data with a limited amount of observational data in continental-scale hydrodynamic models. Data assimilation (DA) is a mathematical technique that combines a physical model with external observations, accounting for their uncertainties, to improve model outputs or replicate the evaluation of an actual

system (Emery et al., 2020a). By leveraging remote sensing data, DA methods can be used to bridge the gap between models and ground observations. DA approaches are widely used in meteorology and oceanography (e.g., Anderson, 2007; Evensen and van Leeuwen, 2002; Miyoshi and Yamane, 2007) and have recently been used in large-scale hydrology (e.g., Clark et al., 2008; Emery et al., 2018; Michailovsky et al., 2013; Paiva et al., 2013a; Revel et al., 2021; Wongchuig et al., 2019). They have also been used to correct hydrodynamic parameters such as river bathymetry (Brêda et al., 2019; Yoon et al., 2012),

Manning's coefficient (Emery et al., 2020a; Pedinotti et al., 2014), and floodplain bathymetry and slope (Durand et al., 2008). Emery et al., (2020c) used DA to improve the accuracy of runoff forcing by integrating discharge observations. Using operation system simulation experiments, researchers have thoroughly investigated the potential for improving river discharge through the assimilation of remote sensing data (Andreadis et al., 2007; Andreadis and Schumann, 2014; Biancamaria et al., 2011; Revel et al., 2019, 2021). In situ (Clark et al., 2008; Paiva et al., 2013a; Wongchuig et al., 2019) or remotely sensed (Emery

et al., 2020b; Feng et al., 2021; Ishitsuka et al., 2020) discharge assimilation performs better, but the unavailability of ground observations and the limitations of remotely sensed river discharge values may hamper the performance of these DA schemes. Thus, DA approaches based on remotely sensed data can be used to improve the performance of global hydrodynamic models. Although DA approaches can improve model performance, hydrodynamic models are not yet mature enough to directly assimilate satellite altimetry data (Emery et al., 2020a). Because of ambiguity in digital elevation models (DEMs), flaws in

hydraulic parameters (e.g., river bathymetry), and the simplification of cross-section parameters, simulated WSE may have substantial errors. Several methods have been used to circumvent these limitations, including assimilating anomalies (i.e., removing the long-term mean WSE) and using a common datum (e.g. Emery et al., 2020a; Michailovsky et al., 2013; Paiva et al., 2013a; Wongchuig-Correa et al., 2020). The inaccuracies that cause biases between simulated WSE and altimetry can be decreased by using anomalies or a common datum when assimilating satellite altimetry into large-scale hydrodynamic models.

To improve river discharge estimation in the Brahmaputra River, Michailovsky et al., (2013) assimilated measurements from the ENVISAT satellite into a rainfall-runoff model using a common reference for satellite altimetry and simulated river depth (i.e., adding the difference between modeled river depth and altimetry elevation to satellite altimetry). Likewise, anomalies from ENVISAT observations were assimilated into a continental-scale hydrologic/hydrodynamic model and compared to insitu and remotely sensed river discharge data in the Amazon Basin (Paiva et al., 2013a). Moreover, global-scale





hydrodynamic modeling studies have used anomaly assimilation to eliminate biases in simulated WSE (Brêda et al., 2019; Emery et al., 2020a; Paiva et al., 2013a; Wongchuig-Correa et al., 2020). However, anomaly assimilation does not provide accurate river discharge estimates for the Amazon Basin (Paiva et al., 2013a), as it cannot compensate for discrepancies in flow dynamics between observations and simulations. These differences in flow dynamics can be attributed to several factors, including differences in amplitude due to limited river width (De Paiva et al., 2013; Yamazaki et al., 2012), differences in

seasonal flow due to failure to capture anthropogenic activity (Hanazaki et al., 2022; Pokhrel et al., 2018; Shin et al., 2020), and differences in flow variation due to the assumption of rectangular cross-sections (Neal et al., 2015; Saleh et al., 2013). Given such uncertainties in parameters and the structural simplification of current hydrodynamic models, anomaly assimilation of satellite altimetry may not be effective for estimating river discharge (Liu et al., 2012; Paiva et al., 2013a). Therefore, alternative approaches to direct and anomaly assimilation are required to integrate satellite altimetry into existing

hydrodynamic models.

In the present study, we evaluated the potential of assimilating satellite altimetry into a global-scale hydrodynamic model to improve river discharge estimation. We investigated methods of assimilating satellite altimetry data into a hydrodynamic model (within current limitations) without contamination from the errors of simulated WSE. Large biases between satellite altimetry and simulated WSE are driven by uncertainties in parameters, whereas simplified physics and a lack of representation

of anthropogenic activity (e.g., reservoir operations) introduce differences in the WSE distribution between simulations and observations. To effectively replace direct value assimilation, we propose alternative methods for DA in the Amazon Basin, including anomaly and normalized value assimilation. The hydrodynamic model used in this study was the Catchment-based Macro-scale Floodplain model (CaMa-Flood: Yamazaki et al., 2011) with the local ensemble transform Kalman filter (LETKF: Hunt et al., 2007), which we used to assimilate satellite altimetry using a physically-based empirical localization approach

(Revel et al., 2019). The methodology is described in Section 2, and the findings are presented in Section 3. The discussion and conclusion are presented in Sections 4 and 5, respectively.

## 2    Methodology

### 2.1    Data assimilation framework

Using a physically-based empirical localization approach, we developed a DA framework to incorporate satellite altimetry

into a hydrodynamic model (Revel et al., 2021). The DA framework developed in this study is represented schematically in Figure 1a. A collection of runoffs created with Earth2Observe's "Global Earth Observation for Integrated Water Resource Assessment" (E2O), a tier-2 Water Resources Reanalysis (WRR2) runoff data set, forced the ensemble simulations. As runoff is the single largest source of error in hydrodynamic modeling (Paiva et al., 2013a; Wongchuig et al., 2019), we simply perturbed the runoff forcing ("Runoff Ensemble"). CaMa-Flood (Yamazaki et al., 2011) was the hydrodynamic core of the DA

scheme, and LETKF (Hunt et al., 2007) was the DA algorithm. CaMa-Flood simulations provide the current water state (i.e., WSE) and correct that value using satellite altimetry. The assimilation scheme takes advantage of physically based empirical local patches (Revel et al., 2019). The initial water state at time $T$ ($x_T^a$) and runoff are used to simulate the forecasted water state at time $T + \Delta T$ ($x_{T+\Delta T}^f$) using the CaMa-Flood hydrodynamic model. The water status is then updated ($x_{T+\Delta T}^a$) via DA, and any modifications are transferred to the initial condition of the following time step. In anomaly and normalized value

assimilation scenarios, the forecasted water state is transformed to anomalies or normalized values using the long-term mean and standard deviation for the assimilation of converted (anomalies or normalized values) satellite altimetry. Then the assimilated water states expressed as anomalies or normalized values were converted into natural values (i.e., corrected WSE). Further information about the transformation of water states is presented in Section 2.3.

To match the WSE obtained from satellite altimetry, we allocated virtual stations (VSs) to the CaMa-Flood river network,

accounting for the Multi-Error-Removed Improved-Terrain DEM (MERIT DEM; Yamazaki et al., 2017, 2019) elevations and river size. The methods used to allocate VSs to the CaMa-Flood river network are illustrated in  Figure 1b. First we digitized to the high-resolution (i.e., 3arc-sec) MERIT Hydro (conditioned DEM) map by using latitude and longitude information to identify the nearest river. The high-resolution locations were then mapped to coarse-resolution river reaches, which were used in CaMa-Flood simulations. Finally, VSs with considerable variation in mean WSE compared to the MERIT Hydro (Yamazaki

et al., 2017, 2019) elevation (expressed as riverbank height) were filtered through comparison of mean observations and



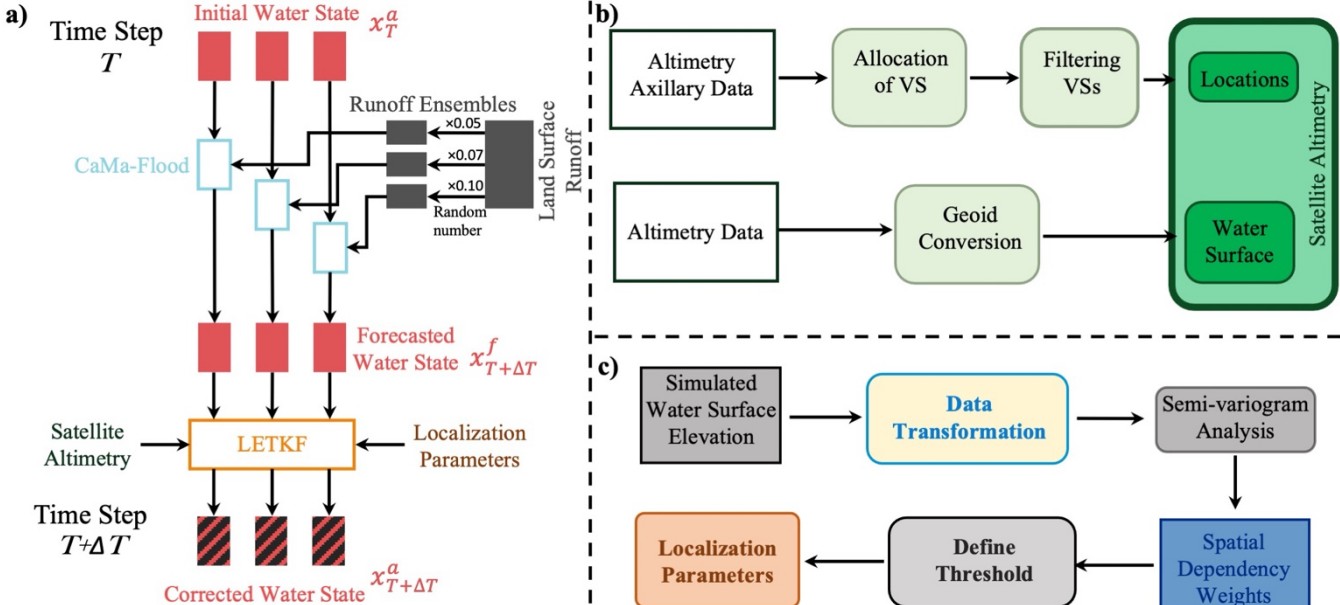

**Figure 1: a) Data assimilation framework, b) schematic diagram of satellite altimetry preprocessing, and c) derivation of the localization parameters.**

riverbank heights (i.e., VSs with mean WSE above or below the 10 m riverbank height of the MERIT river network were removed). Next all satellite altimetry elevations were converted into EGM96 from EGM08 via geoid conversion. Allocation of VSs to the CaMa-Flood river network is a vital step in the assimilation framework.

Using simulated long-term WSE values, we determined the localization parameters (i.e., local patch and observation
localization weights; Figure 1c). Deriving empirical localization parameters involved simulating WSE with CaMa-Flood, processing the data, running semi-variogram analyses, and assigning a threshold to spatial dependence weights. The physically-based empirical localization DA approach outperformed traditional localization methods (Revel et al., 2019). Hence, when combined with LETKF, these localization parameters provide a foundation for efficient continental-scale DA.

## 2.2 Hydrodynamic Model

To diagnose the time-varying water states in the DA scheme, we used CaMa-Flood (Yamazaki et al., 2011), which is a large-scale distributed hydrodynamic model. CaMa-Flood uses a local inertial flow equation, which is a computationally efficient variant of the shallow-water equation (Bates et al., 2010; Yamazaki et al., 2011), to determine river hydrodynamics (e.g., discharge, WSE, flood depth, flooded area). Runoff (surface and subsurface flow of water per unit area) from a land surface model (LSM) forces the model, and water is routed through the river network at adaptive time steps (Yamazaki et al., 2013).
CaMa-Flood is capable of simulating floodplain dynamics, complex hydrodynamics such as the backwater effect (Yamazaki et al., 2011, 2012), and bifurcation flow (Yamazaki et al., 2014b). It is a physically-based model that can simulate WSE; combining CaMa-Flood with MERIT DEM (Yamazaki et al., 2017, 2019) improves its performance relative to satellite altimetry. Consequently, the CaMa-Flood hydrodynamic model is appropriate for the DA framework described in Section 2.1. We used CaMa-Flood version 4.0, which was developed with MERIT-DEM and MERIT-Hydro (Yamazaki et al., 2017, 2019)
at a spatial resolution of 0.1°. The simulations used the standard parameters (river channel depth, river width, roughness coefficient, and floodplain profile) of CaMa-Flood. The river channel depth was estimated using a power law (Yamazaki et al., 2011; Zhou et al., 2022). River widths were determined with remote sensing (Yamazaki et al., 2014a), and the roughness coefficient was approximated as a global constant (0.03). MERIT-DEM and MERIT-Hydro were used to construct the river network (Yamazaki et al., 2017, 2019).



## 2.3    Water surface elevation transformation

Because of the large biases in simulated WSE, direct comparison with satellite altimetry is difficult. Figure 2a presents an example of WSE bias and a comparison of satellite altimetry with simulated WSE. This figure shows that direct DA can introduce additional biases into assimilated WSE. These biases are caused by inaccuracies in parameters such as riverbank elevation height errors and river bathymetry errors as well as differences in elevation due to hydrodynamic model resolution (i.e., models assume the unit-catchment outlet elevation as the riverbank elevation of the river reach). Converting WSE into anomalies can reduce the challenges created by large differences between simulated and observed WSE values. WSE anomalies were generated by subtracting the time-averaged reference WSE (i.e., the long-term mean) from the current WSE. Therefore, each ensemble member had a different reference WSE value.

Although the use of anomalies can overcome the bias between observations and simulations, differences in flow variation between simulated and observed WSE remain (e.g., a difference in the amplitude of WSE variation, upstream water regulations that do not represent in the model). An example of a difference in flow variation is presented in Figure 2b. The flow dynamic variation between simulations and observations can be overcome by using normalized values (i.e., subtracting the long-term mean and dividing by standard deviation) when assimilating satellite altimetry into contemporary hydrodynamic models. To

**Figure 2: a) Schematic diagram of the bias between simulated and observed WSE and an example of the bias between simulated and observed water surface elevation (WSE) at the HydroWeb VS R_AMAZONAS_JARI_KM0529. b) Difference between simulation (CaMa-Flood) results and observations with distribution differences shown as a boxplot, probability distribution, and an example of difference in amplitude between simulated and observed WSE at the HydroWeb VS R_AMAZONAS_JUTAI_KM3182.**



more accurately compensate for distribution discrepancies and estimate river discharge, we used normalized value assimilation.
We normalized the current WSE values using the time-averaged reference WSE and standard deviation of WSE for each perturbation in the ensemble. Hence, each perturbation had a unique reference WSE and standard deviation of WSE.

## 2.4 Local Ensemble Transformation Kalman Filter

DA aims to overcome differences between observations and simulations by combining uncertain and complementary
information from observations. In this study, LETKF (Hunt et al., 2007), which is a computationally efficient variant of the
ensemble Kalman filter (EnKF: Evensen, 2003), an advanced Kalman filter (Kalman, 1960), was used as the DA method. We used a physically-based empirical localization approach (Revel et al., 2019, 2021) to enhance the computational efficiency of global-scale DA.
The LETKF is a commonly used DA algorithm (e.g., Feng et al., 2021; Ishitsuka et al., 2020; Revel et al., 2019, 2021b; Wongchuig-Correa et al., 2020) for nonlinear models, which are needed for modeling hydrodynamic processes. The nonlinear
hydrodynamic model can be shown in discrete form as follows:

$$x_{k+1} = \mathcal{M}(x_k, u_k, \vartheta) + q_k, \tag{1}$$

where $x$, $u$, and $\vartheta$ represent the vector of the state variable, model forcing, and model parameters, respectively. The nonlinear model operator, $\mathcal{M}$, is related to the time interval of $t_k$ to $t_{k+1}$, whereas errors in the model structure, parameters, forcing, and antecedent states are represented by $q_k$. All state variables in CaMa-Flood, such as river discharge, WSE, flooded area, flood height, and storage, are included within the vector $x$. The model states can be related to the observations as follows:

$$y_k = H(x_k) + \varepsilon_k, \tag{2}$$

where $y$ is the observation vector; $\varepsilon$ is the vector of observation errors; and $H$ is the linear observation operator, which relates the model states ($x$) to the observations ($y$). In this study, the observations were WSE obtained from satellite altimetry. In the anomaly and normalized value assimilations, the observed and forecasted states were transformed into anomalies and normalized values, respectively (Section 2.3, Figure 2). The LETKF assimilation algorithm was used to obtain the optimal estimate of the model state variable $X^a$ (analysis) considering the model and observation errors. LETKF analysis is expressed
as

$$X^a = X^f + E^f \left[ VD^{-1}V^T(HE^f)^T \left(\frac{R}{w}\right)^{-1} (Y^o - HX^f) + \sqrt{m-1}VD^{-1/2}V^T \right], \tag{3}$$

where $X^a$ is the posterior state estimator (or analysis), $X^f$ is the prior state estimator (or forecast), $Y^o$ is the observation (i.e., the WSE value obtained from satellite altimetry), $H$ is the observation operator, $m$ is the ensemble size, $E^f$ is the prior state error covariance obtained directly from the perturbations, $R$ is the observation error covariance determined from the uncertainty of the measurements, $w$ is the weighting term for observation localization calculated with semi-variogram
analysis of the simulated WSE (Revel et al., 2019), and $VDV^T$ is defined as

$$VDV^T = (m-1)I + (HE^f)^T R^{-1}HE^f \tag{4}$$

where $I$ is the unit matrix of dimension $m \times m$, representing the number of perturbations. $VD^{-1}V^T$ and $VD^{-1/2}V^T$ are calculated through eigenvalue decomposition of $VDV^T$.

## 2.5 Generation of Ensembles

CaMa-Flood diagnoses terrestrial water dynamics forced by surface and subsurface runoff values simulated using LSMs. LSMs
are subject to flaws such as simplified physics (e.g., hill slope dynamics; (Fan et al., 2019) and uncertainty in the forcing data). We stochastically perturbed the runoff forcing of the CaMa-Flood hydrodynamic model. The errors in runoff can be attributed





to the limitations of LSM physics and uncertainty in the forcing data sets. We assumed that the uncertainty of the LSM's physical processes could be represented by the variability of the multi-model runoff diagnosed in the E2O WRR2 data set (Dutra et al., 2017). The uncertainty of the forcing (e.g., precipitation, radiation) was represented by normally distributed random numbers with a standard deviation calculated from the ensemble spread of the 20th-century atmospheric model ensemble (ERA20CM: Hersbach et al., 2015) runoff data set. We multiplied each runoff from the E2O WRR2 data set (seven runoff data sets from E2O WRR2 were used) by a random number from a normal distribution with mean = 1 and standard deviation = 0.1 according to the ERA20CM runoff ensemble. We used seven runoff outputs from the models HTESSEL (Balsamo et al., 2011), PCR-GLOBWB (Van Beek et al., 2011; Sutanudjaja et al., 2014), JULES (Best et al., 2011; Clark et al., 2011), LISFLOOD (Burek et al., 2013; Van Der Knijff et al., 2008), ORCHIDEE (d'Orgeval et al., 2008), WaterGAP3 (Flörke et al., 2013; Verzano, 2009), and W3 (Van Dijk et al., 2013) of E2O WRR2 (Dutra et al., 2017). Therefore, 49 perturbations were used to prepare the "Runoff Ensembles" (Figure 1) using runoff fields from the E2O WRR2 runoff product, which generally produce a reasonable ensemble of runoff forcing for global DA.

## 2.6 Experimental Design

We performed three types of experiment: direct DA (Exp 1), anomaly DA (Exp 2), and normalized DA (Exp 3). Exp 2 was performed because assimilating WSE anomalies rather than direct values can overcome the errors associated with direct DA. Although anomalies can overcome the biases between observations and simulations, differences in flow variation between simulated and observed WSE could not be overcome by anomaly DA method (Figure 2b). To overcome the flow dynamic variation between simulations and observations, we performed Exp 3. Both forecasted values and observations were transformed into anomalies (Exp 2) and normalized values (Exp 3) for the DA experiments. The three assimilation approaches were used to identify the optimal assimilation methodology for improving discharge estimation within the present limits of hydrodynamic modeling. The anomalies and normalized values were calculated from the long-term (2000–2014) mean and standard deviation of WSE for the anomaly and normalized value DA experiments. For all experiments, simulations began on January 1, 2009, and ran through December 31, 2014. The year 2008 was used for spin-up.

We selected the Amazon Basin as the test area for our DA experiments. The Amazon Basin is the world's largest hydrological system, with a catchment area of approximately 6 million km$^2$ (Reis et al., 2019), and contributes nearly one fifth of the total fresh water discharged into the ocean (Paiva et al., 2013a). The flow dynamics of the Amazon Basin, ranging from seasonal flooding (Papa et al., 2010; Prigent et al., 2020) to complex river hydraulics such as hysteresis in the stage-discharge relationship driven by the backwater effect (Paiva et al., 2013b; De Paiva et al., 2013), have been studied extensively. This basin receives substantial annual rainfall ($\approx 2200\ mm$) with high spatial heterogeneity and experiences distinct rainy and dry seasons in the southern and eastern portions. The major advantage of analyzing the Amazon Basin is the availability of a large number of observations (Fassoni-Andrade et al., 2021).

## 2.7 Observations

### 2.7.1 Satellite altimetry

We used satellite altimetry as observations for all DA experiments (Section 2.6). Satellite altimetry was originally developed to observe ocean surfaces, but its application has expanded through the creation of algorithms to detect surface water dynamics (Birkett et al., 2002; Crétaux et al., 2009; Santos da Silva et al., 2010). Satellite altimetry data were obtained from HydroWeb (https://hydroweb.theia-land.fr/). Satellite altimetry measurements from ENVISAT and Jason 1, and Jason 2 were used depending on data availability during the simulation period (2009–2014), as listed in Table 1. Table 1 summarizes the availability periods, temporal resolutions, cross-track distances, and measurement errors of the satellites used in this study. The spatial distribution of VSs is illustrated in Figure 3. Using the methodology described in Section 2.1, we allocated the VSs to river pixels in CaMa-Flood (Figure 1b). Preprocessing excluded around 3% of VSs from analyses, which may have generated considerable inaccuracies, in particular in the experiments with direct value assimilation (Exp 1a and Exp 1b). The WSE data obtained from satellite altimetry were converted from EGM08 to EGM96, as the EGM96 geoid model is used in MERIT-DEM/MERIT-Hydro (Yamazaki et al., 2017, 2019).



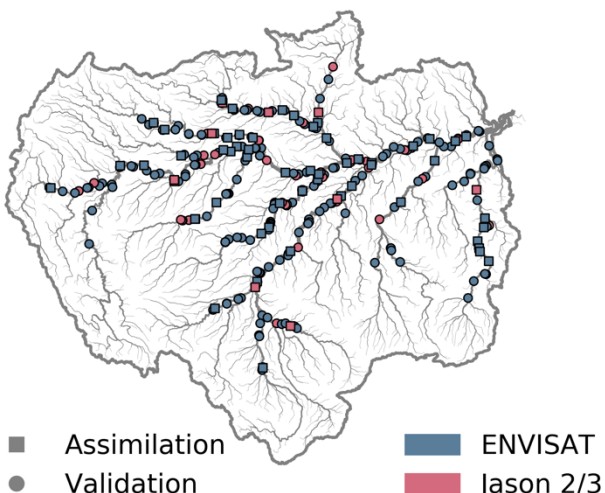

**Figure 3: Spatial distribution of satellite virtual stations (VSs) used in this study. ENVISAT VSs are shown in red, and Jason 1 and Jason 2 VSs are in blue. VSs used for assimilation and validation are indicated with squares and circles, respectively.**

**Table 1: Summary of satellite altimetry data used in this study. Period of availability, measurement error, temporal resolution, and cross-track distance are shown.**

| Satellite | Period | Measurement Error (mm) | Temporal Resolution (days) | Cross-Track Distance (km) |
|---|---|---|---|---|
| ENVISAT | 2002–2012 | 35 | 30–35 | 80 |
| Jason 1 | 2001–2013 | 28 | 10 | 315 |
| Jason 2 | 2008–present | 28 | 10 | 315 |

255 **2.7.2 Validation data**

As we assimilated only WSEs from satellite altimetry, we used Global Runoff Data Centre (GRDC) river discharge data for validation. We used river discharge gauges located in the main river reaches (upstream catchment area > 1000 km2) and gauges with observational data covering at least 1 year of the simulation period. Furthermore, we randomly segregated the satellite altimetry measurements from 324 VSs into assimilation (80%) and validation (20%) data sets (Figure 3). Consequently, we
260 used only 80% of the VSs for the DA experiments.

**2.8 Evaluation Diagnostics**

We evaluated relative assimilation efficiency using several diagnostics. The difference in correlation coefficient ($\Delta r$) between assimilated and open-loop simulations was assessed to evaluate improvement in the flow pattern of the discharge. $\Delta r$ was calculated as

$$\Delta r = r_{asm} - r_{opn}, \tag{5}$$

265 where the correlation coefficients of the assimilated and open-loop simulations are represented by $r_{asm}$ and $r_{opn}$, respectively. Then the relative efficacy of WSE was assessed with the relative root mean square error ($rRMSE$):



$$rRMSE = RMSE_{asm} - RMSE_{opn} \tag{6}$$

$$RMSE = \sqrt{\frac{\sum_{i=1}^{N}(s_i - o_i)^2}{N}} \tag{7}$$

where $RMSE_{asm}$ and $RMSE_{opn}$ are the $RMSE$ values of the assimilated and open-loop simulations, respectively. $s$ and $o$ are simulation results and observations, respectively. $N$ is the number of observations in the timeseries. The Nash-Sutcliffe (Nash and Sutcliffe, 1970) efficiency-based assimilation index ($NSEAI$; Revel et al., 2021) was used to evaluate the improvement in river discharge with DA:

$$NSEAI = \frac{NSE_{asm} - NSE_{opn}}{1 - NSE_{opn}} \tag{8}$$

where $NSE_{asm}$ and $NSE_{opn}$ are Nash-Sutcliffe (Nash and Sutcliffe, 1970) efficiencies for the assimilated and open-loop simulations, respectively. Similarly, the Kling-Gupta (KGE: Kling and Gupta, 2009) efficiency-based assimilation index ($KGEAI$) was used to evaluate improvement. The relative interval skill score ($rISS$) was used to compare the ensemble spread of the assimilated and open-loop simulations. $rISS$ is defined as follows:

$$rISS = \frac{(ISS_{asm} - ISS_{opn})}{ISS_{opn}} \tag{9}$$

$$ISS_\alpha = \sum_{i=1}^{N} iss_\alpha(l_i, u_i, o_i) \tag{10}$$

$$iss_\alpha(l, u, o) = \begin{cases} (u - l)\,; if\ l < o < u \\ (u - l) + \frac{2}{\alpha}(l - o)\,\ ; if\ o < l \\ (u - l) + \frac{2}{\alpha}(o - u)\,; if\ o > u \end{cases} \tag{11}$$

where $ISS_{asm}$ and $ISS_{opn}$ are the $ISS$ values (Gneiting and Raftery, 2007) of the assimilated and open-loop simulations, respectively. $u$ and $l$ are the upper and lower confidence intervals for the estimate, $o$ is the observed value, and $\alpha$ is the significance level. By rewarding narrow confidence intervals and penalizing observations outside the nominal confidence intervals, the $ISS$ incorporates both sharpness (i.e., the size of the confidence interval) and reliability (i.e., the proportion of observations that fall within the nominal confidence interval specified). When balancing sharpness and reliability, a relative comparison of $ISS$ ($rISS$) allows for the evaluation of ensemble models; those with better performance have lower $ISS$ values. The significance level ($\alpha$) was set to 0.05 in this study. Furthermore, relative sharpness ($rSharpness$) and difference in reliability ($\Delta Reliability$) were used to evaluate relative assimilation performance.

We used $NSE$ (Nash and Sutcliffe, 1970) and KGE (Kling and Gupta, 2009) to evaluate the overall performance of river discharge. Furthermore, $RMSE$, bias between the means of observations and simulation results ($BIAS$), and the difference in amplitude ($\Delta A$) of WSE were evaluated.

## 3    Results

### 3.1    Relative Performance Evaluation

In this section, we present the relative performance of each assimilation approach, namely, the direct, anomaly, and normalized value DA experiments. Here we analyze the performance of assimilated values with respect to the open-loop simulation. $\Delta r$ represents the relative change in $r$ between the open-loop or assimilation results and observations. $rRMSE$ represents the deviation in the $RMSE$ of assimilation relative to that of open-loop simulation. $rISS$, $rSharpness$, and $\Delta Reliability$ are used



to assess changes in ensemble spread between the open-loop and assimilated simulations. The results are presented in the order
direct (Exp 1), anomaly (Exp 2), and normalized value (Exp 3) assimilation, followed by a comparison of the relative performance (i.e., $\Delta r$, $NSEAI$, $KEGAI$, $rISS$, $rSharpness$, and $\Delta Reliability$) of these experiments.

### 3.1.1    Direct assimilation of satellite altimetry

In Exp 1, we assimilated direct satellite altimetry measurements into the CaMa-Flood hydrodynamic model. Figure 4a shows the improvement (degradation) in the correlation coefficient in green (violet) for river discharge at the GRDC locations in this
study. $r$ improved in 8.1% of GRDC locations out of the 86 used for evaluating river discharge, whereas half of the gauges (48.8%) showed no difference. $r$ was reduced ($\Delta r < 0$) at several locations along the Madeira, Negro, and Purus tributaries (accounting for 43.0% of total gauges). We evaluated WSE performance using the relative change in $RMSE$ between observed and simulated discharge ($rRMSE$). Large negative (positive) values for $rRMSE$ indicate better (worse) performance of the DA scheme, which is denoted by blue (red). Overall, 56.4% and 50.8% of assimilation and validation VSs, respectively,
showed reductions in $RMSE$ with the assimilation of satellite altimetry into a hydrodynamic model. The WSE estimates obtained from the assimilated simulation with direct DA into a model were degraded (assimilation: 39.0%, validation: 41.5%) relative to the open-loop simulation in the Amazon mainstem and the Negro, Branco, Madeira, and Xingu rivers (Figure 4b). A limited number of gauges demonstrated no change with direct DA.

Figure 4c, d, and e depicts hydrographs at Labera in the Purus River, Santos Dumont in the Purus River, and Santo Antonio
Do Ica in the Amazon River, in that order. Each panel shows observations (black line), open-loop simulation results (blue line), assimilated discharge (orange line), and 95% confidence bounds for assimilated and open-loop river discharge. The discharge at Labera station (Figure 4c) improved in terms of $NSE$ and $ISS$ but not $r$. Substantial improvement in the 95% ensemble spread was evident until mid-2010, when the ENVISAT satellite was available. However, confidence intervals became larger after 2010. DA marginally improved $NSE$ scores, with low flows well replicated but peak flows showing some fluctuations.
Santos Dumont (Figure 4d) showed an improvement in the correlation coefficient of river discharge, although NSE suffered from substantial underestimation of high flow. $ISS$ increased by 29%, primarily because of an improvement in sharpness, but reliability decreased. Figure 4e illustrates the variation in discharge at a station located in the mainstem of the Amazon River (Santo Antonio Do Ica), showing an improvement in $NSE$ values but a weakening of the correlation coefficient. At this location, the tradeoff between reliability and sharpness is strong. Sharpness is often enhanced when direct satellite altimetry
measurements are assimilated into an uncalibrated hydrodynamic model, but reliability is reduced.

In summary, direct DA generally improved flow dynamic simulation to a moderate extent. When direct satellite altimetry measurements were assimilated into the hydrodynamic model, the sharpness of river discharge improved. Furthermore, the accuracy of WSE estimates also improved with DA.

### 3.1.2    Anomaly assimilation of satellite altimetry

In Exp 2, anomalies of satellite altimetry were assimilated to anomalies of simulated WSE, with both anomalies produced using long-term means (for VS: mean of the available period; for WSE simulation: 2000–2014). Figure 5a depicts the $\Delta r$ of river discharge in Exp 2, with green indicating an improved $r$ relative to the open-loop simulation, which accounted for around 53.5% of GRDC gauges used to evaluate river discharge. Some degradation (purple) in $r$ was observed in the Madeira and Purus rivers, Amazon mainstem, and smaller river reaches. WSE estimates improved in 76.1 % of assimilation VSs and 80.0 %
of validation VSs (Figure 5b). WSE decreased (in terms of $RMSE$) in the Jurua and Purus rivers, although nearly all other river reaches showed increases in the accuracy of WSE calculations with DA.



Figure 5c–e displays hydrographs of the Jurua (Gaviao), Amazon (Manacapuru), and Negro (Serrinha) rivers, respectively. At Gaviao station (Figure 5c), the $r$ of river discharge increased slightly, whereas $NSEAI$ and $rISS$ decreased. Although low flows were adequately characterized during the brief observation period, peaks were exaggerated, resulting in low $NSE$ and

high $ISS$ for the assimilated simulation. River discharge in the Amazon mainstem, notably at Manacapuru gauge (Figure 5d), was well characterized, with improvements in $\Delta r$ and $rISS$ but deterioration of $NSE$ values. By contrast, the flow variation was accurately defined at Serrinha station in the Negro River (Figure 5e) with anomaly DA and an uncalibrated hydrodynamic model ($\Delta r$, $NSEAI$, and $rISS$ were improved). Through anomaly assimilation into an uncalibrated hydrodynamic model, the flow dynamics (characterized by $r$) of the Amazon Basin improved, although $NSE$ and $ISS$ values worsened slightly.

**Figure 4: a)** Difference in the correlation coefficient of river discharge ($\Delta r$) and **b)** relative Root Mean Square Error ($rRMSE$) of water surface elevation for Exp 1. Circles indicate virtual stations used for data assimilation, and squares are virtual stations used for validation on the WSE plots. Hydrographs recorded at Labera on the Purus River, Santos Dumont on the Jurua River, and Santo Antonio Do Ica on the Amazon River are presented in panels c, d, and e, respectively. The locations of the hydrographs shown in panels c, d, and e are presented in panel a. Discharge observations are shown in black, assimilated simulation results in orange, and open-loop simulation results in blue. The color range indicates the 95% confidence interval used to calculate the relative interval skill score ($rISS$). $\Delta r$, Nash-Sutcliffe efficiency-based assimilation index ($NSEAI$), and $rISS$ are shown at the bottom of each panel.





**Figure 5: a) Difference in the correlation coefficient of river discharge ($\Delta r$) and b) relative root mean square error ($rRMSE$) of water surface elevation for Exp 2. Circles indicate virtual stations used for data assimilation, and squares are virtual stations used for validation on the WSE plots. Hydrographs recorded at Manicore on the Madeira River, Aruma on the Purus River, and Sao Felipe on the Negro River are presented on panels c, d, and e, respectively. The locations of the hydrographs shown in panels c, d, and e are presented in panel a. Discharge observations are shown in black, assimilated simulation results in orange, and open-loop simulation results in blue. The color range indicates the 95% confidence interval used to calculate the relative interval skill score ($rISS$). $\Delta r$, Nash-Sutcliffe efficiency-based assimilation index ($NSEAI$), and $rISS$ are shown at the bottom of each panel.**

Overall, the discharge estimates improved moderately with the assimilation of WSE anomalies into the hydrodynamic model. The seasonality of river discharge improved considerably in most river reaches with anomaly assimilation. Furthermore, WSE calculation was improved in many Amazon Basin river reaches.

### 3.1.3    Normalized assimilation of satellite altimetry

In Exp 3, we assimilated normalized values of WSE. Long-term statistics (mean and standard deviation of WSE for 2000–2014) were used to generate normalized values of DA for Exp 3a. Figure 6a and b represent the $\Delta r$ of river discharge and $rRMSE$ of WSE, respectively, for Exp 3. A total of 60.5% of GRDC gauges demonstrated a positive $\Delta r$, whereas decreases



were evident in the Purus and Madeira rivers as well as the Amazon mainstem. A considerable number of VSs showed an improvement in WSE calculations with the normalized DA technique (85.6% for both assimilation and validation VSs; Figure 6b).


The lower panels of Figure 6 illustrate flow dynamics along the Amazon mainstem (Sao Paulo De Olivenca; Figure 6c) and Japura (Vila Bittencourt; Figure 6d) and Negro (Curicuriari; Figure 6e) rivers. The discharge at Sao Paulo De Olivenca station in the Amazon mainstem (Figure 6d) resembled the observed river discharge. Although $NSEAI$ and $rISS$ were both enhanced, $\Delta r$ was marginally degraded. Note that normalized value DA replicated the flow dynamics of the observations well,


showing a secondary peak (e.g., October 2009) at the Sao Paulo De Olivenca station that was absent in the open-loop simulation. Although low flows and other fluctuations were accurately portrayed along the Japura River (Vila Bittencourt; Figure 6d), the relative assimilation efficiency metrics had low values. Figure 6e illustrates a hydrograph of the Curicuriari gauge along the

**Figure 6: a) Difference in the correlation coefficient of river discharge ($\Delta r$) and b) relative root mean square error ($rRMSE$) of water surface elevation for Exp 3. Circles indicate virtual stations used for data assimilation, and squares are virtual stations used for validation on the WSE plots. Hydrographs recorded at Humaita on the Madeira River, Santos Dumont on the Jurua River, and Canutama on the Purus River are presented on panels c, d, and e, respectively. The locations of the hydrographs shown in panels c, d, and e are presented in panel a. Discharge observations are in black, assimilated simulation results in orange, and open-loop simulation results in blue. The color range indicates the 95% confidence interval used to calculate relative interval skill score ($rISS$). $\Delta r$, Nash-Sutcliffe efficiency-based assimilation index ($NSEAI$), and $rISS$ are shown at the bottom of each panel.**




Negro River. The discharge at Curicuriari was well characterized, with a positive $\Delta r$ and $NSEAI$ and a negative $rISS$.
Normalized DA using an uncalibrated hydrodynamic model improved the characterization of river discharge in terms of
seasonal dynamics, overall accuracy, and the tradeoff between sharpness and reliability.
The normalized DA approach improves flow variation in most river reaches. In the normalized assimilation experiment, WSE
estimates improved in most Amazon Basin river reaches.

### 3.1.4    Comparison of assimilation experiments

To evaluate the relative improvement associated with DA, we evaluated only those GRDC gauges located in river reaches
observed through satellite altimetry (satellite coverage; Figure S1). The effectiveness of assimilation for GRDC gauges located
outside the area of the satellite observations is poor, with very little difference between open-loop and assimilated simulations.
Approximately 75% of the 86 GRDC gauges lay outside the satellite coverage area (Figure S1). The 21 gauges within the
satellite coverage area were used to assess the relative improvement among experiments. Table 2 presents median relative
performance statistics for river discharge estimates for all experiments. Positive values for $\Delta r$, $NSEAI$, $KGEAI$, and
$\Delta Reliability$ indicate that DA improved river discharge estimation. Negative values for $rISS$ and $rSharpness$, in contrast,
demonstrate improvement in river discharge estimation with DA. For all experiments, Figure 7a displays the kernel density
estimate of the probability density function for the $\Delta r$ of the river discharge. All experiments except Exp 1 showed
improvement in the median $\Delta r$ ($> 0$), demonstrating improvement in the flow regime with DA for at least 50% of gauges.
The $\Delta r$ for river discharge in Exp 1 showed a left-skewed distribution, which suggests deterioration in seasonality at many
gauges (75%). Approximately 70% of gauges showed improvements in flow regime characterization in the anomaly and
normalized value DA experiments. However, only Exp 3 had a positive median $NSEAI$, which indicates that at least 50% of

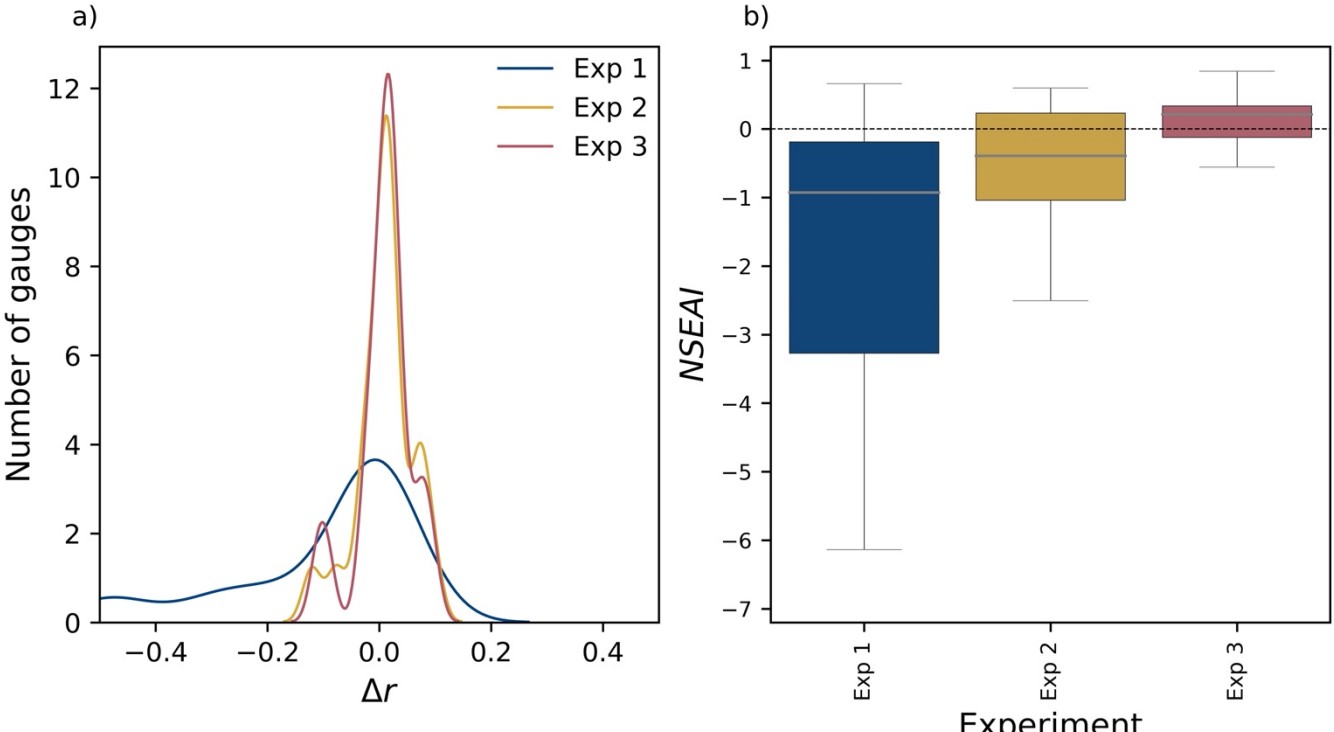

**Figure 7: a) Cumulative distribution of the correlation coefficient ($\Delta r$) for each experiment, shown in blue, yellow, and red for direct (Exp 1), anomaly (Exp 2), and normalized value (Exp 3), respectively. b) Boxplots of the Nash-Sutcliffe based assimilation index ($NSEAI$) of assimilated compared to open-loop discharge for all the experiments. Boxes in blue, yellow, and red indicates direct (Exp 1), anomaly (Exp 2), and normalized value (Exp 3), respectively.**





gauges had improved *NSE* values with normalized value DA (Table 2). Figure 7b shows boxplots of *NSEAI* for all experiments, demonstrating considerable improvement in the DA experiments with normalized value assimilation. *KGEAI* followed the same pattern, with positive median values for Exp 3.

Compared to $\Delta r$, *NSEAI*, and *KGEAI*, *rISS* showed the opposite trend (Figure S2a), with major improvements resulting from direct DA. A negative *rISS* means that the *ISS* of assimilated discharge was improved, as a lower *ISS* indicates better performance with DA. Direct assimilation (Exp 1) led to a lower median *rISS* value (–0.36), whereas both anomaly and normalized value assimilation had values –0.13 and –0.18, respectively. When evaluating *rISS*, one must consider changes in sharpness (i.e., the width of the confidence interval) and reliability (i.e., the percentage of observations that fall within the

predicted nominal confidence interval; (Michailovsky et al., 2013). A large reduction in sharpness was observed in the direct assimilation experiment (Exp 1), mainly because the assimilation was conducted directly (Figure 4c–e). The lowest reliability reduction was obtained in the normalized value assimilation experiment (Exp 3). The reliability of direct assimilation was reduced by 54%, whereas sharpness improved by 79% in Exp 1 compared to the open-loop simulation (i.e., the 95% confidence interval was larger in the open-loop simulation; Figure S2b–c). Although the confidence bounds (i.e., *sharpness*) were

narrower with direct DA compared to the anomaly and normalized value DA experiments, reliability was degraded by more than 50%.

In summary, considering the improvements measured using multiple evaluation metrics (e.g., *NSEAI*, *KGEAI*), normalized value assimilation (Exp 3) showed the greatest improvement relative to the open-loop simulation, whereas the smallest improvement was obtained from the direct DA experiment (Exp 1). However, the tradeoff between sharpness and reliability

was better in the direct DA experiment, as the assimilations were performed directly. Sharpness was substantially improved in Exp 1. In anomaly and normalized value assimilations, WSE space is affected by the calculation of anomalies or normalized values. Hence, given the current condition of hydrodynamic modeling (i.e., the limitations of hydrodynamic models), normalized value assimilation performed best.

**Table 2: Summary of relative DA efficiency statistics for river discharge in each experiment. The difference in the correlation coefficient ($\Delta r$), Nash-Sutcliffe efficiency-based assimilation index (*NSEAI*), Kling-Gupta efficiency-based assimilation index (*KGEAI*), relative interval skill score (*rISS*), relative sharpness (*rSharpness*), and difference in reliability ($\Delta Reliability$) are shown. Positive values for $\Delta r$, *NSEAI*, *KGEAI*, and reliability represent better performance of DA, and lower values for *rISS* and sharpness indicate improvements due to DA. Improvements in each relative performance metric with DA are highlighted in bold**

| Experiment | $\Delta r$ | *NSEAI* | *KGEAI* | *rISS* | *rSharpness* | $\Delta Reliability$ |
|---|---|---|---|---|---|---|
| Exp 1 | -0.02 | -0.93 | -0.72 | **-0.36** | **-0.79** | -0.54 |
| Exp 2 | **0.01** | -0.39 | -0.39 | **-0.13** | **-0.14** | -0.01 |
| Exp 3 | **0.01** | **0.21** | **0.02** | **-0.18** | **-0.15** | 0.00 |


## 3.2 Absolute Performance Evaluation

In this section, we explore the absolute performance of river discharge and WSE. When analyzing absolute performance, we consider the $r$, *NSE* (Nash and Sutcliffe, 1970), and *KGE* (Gupta et al., 2009) values for river discharge; *RMSE*, *BIAS*, and $\Delta A$ are used for WSE. $r$ is used to assess the seasonality of river discharge estimates, whereas *NSE* and *KGE* are used to

evaluate the overall performance of river discharge estimation. *RMSE* is used to evaluate the overall error of WSE estimation against satellite altimetry observations. Long-term bias is assessed with *BIAS*, and the difference in amplitude between simulated and observed WSE is examined using $\Delta A$. 3.2.1The absolute performance of daily discharge estimates is presented in Section 3.2.1, and the absolute performance of WSE estimation is described in Section 3.2.2.





### 3.2.1    Estimation of daily river discharge

We used $r$, $NSE$, $KGE$, and $Sharpness$ to evaluate daily assimilated river discharge across all experiments, and Table 3 presents the median statistics for each metric. We obtained the reported median values using all GRDC gauges in the Amazon Basin (all) and, more conservatively, using river reaches with satellite altimetry observations (satellite coverage reaches) so the impact of assimilation on river reaches outside the satellite observation area was minimal. In general, the median performance metrics in satellite coverage river reaches were better than the median performance of all discharge gauges,

whereas median $Sharpness$ was worse in satellite coverage river reaches. $Sharpness$ was determined with the average confidence bounds, and river reaches with satellite coverage have high discharge, resulting in larger confidence intervals. Consequently, the median $Sharpness$ estimate for river reaches with satellite coverage was inevitably large. When only river reaches with satellite observations were considered, the $NSE$ of Exp 1 was reduced. Flow patterns improved with the shift from direct to anomaly or normalized value DA. However, the differences between the anomaly and normalized DA

**Figure 8: Performance of daily discharge in terms of the correlation coefficient ($r$), Nash-Sutcliffe efficiency ($NSE$), and Kling-Gupta efficiency ($KGE$) for a) Exp 1, b) Exp 2, and c) Exp 3.**



experiments were marginal. Median $NSE$ and $KGE$ values increased in the order of direct, anomaly, and normalized value DA experiments. However, the direct DA experiments efficiently improved $Sharpness$, thereby increasing confidence in the assimilated river discharge. Exp 1 had the lowest sharpness values for both the entire river and satellite-covered reaches.

Figure 8 shows the spatial distributions of absolute performance metrics (e.g., $r$, $NSE$, and $KGE$) for daily river discharge in all DA experiments (Exp 1, Exp 2, and Exp 3). Figure 8a depicts the spatial distribution of the absolute performance of river

discharge estimates obtained in the direct DA experiment (Exp 1). The $r$ of river discharge estimation for several GRDC gauges was $> 0.8$ (approximately 38%), with a mean $r$ of 0.73. $NSE$ and $KGE$ were $> 0.6$ in 22% and 34% of gauges, respectively, and the median $NSE$ and $KGE$ were 0.13 and 0.46, respectively. Some gauges along the Negro, Jurua, and Upper Solimoes rivers had low accuracy for estimating river discharge through direct DA (Exp 1).

The spatial distribution of the absolute performance of river discharge estimation through anomaly DA is shown in Figure 8b.

Anomaly DA (Exp 2) produced $r$ values of river discharge that were $> 0.8$ in 84% of gauges, with a median $r = 0.84$. In the Amazon Basin, overall river discharge was well characterized by anomaly DA (35% of stations $NSE > 0.6$ and 48% of stations $KGE > 0.6$).

Figure 8c illustrates the performance of Exp 3, showing better performance in large river reaches ($catchment\ area >$ $1000\ km^2$). Nearly 57% of GRDC gauges had $r > 0.8$, with a median $r$ of 0.83 in Exp 3. The preponderance of gauges (76%)

had $NSE > 0.6$, with a median $NSE$ of 0.47. $KGE$ values were greater than 0.6 for 92% of gauges, with a median of 0.62. Most gauges along the Amazon mainstem and Negro, Purus, Madeira, and Jurua rivers reliably estimated river discharge with assimilation of satellite altimetry using the normalized value DA method.

**Table 3: Median performance metrics for daily discharge estimates obtained from DA experiments. Median values for the correlation coefficient ($r$), Nash-Sutcliffe efficiency ($NSE$), Kling-Gupta efficiency ($KGE$), and width of the confidence interval ($Sharpness$) are presented for all GRDC gauges and gauges in the satellite coverage area.**

| Experiment | All | | | | Satellite Converge Reaches | | | |
|:---:|:---:|:---:|:---:|:---:|:---:|:---:|:---:|:---:|
| | $r$ | $NSE$ | $KGE$ | $Sharpness$ ($10^6$) | $r$ | $NSE$ | $KGE$ | $Sharpness$ ($10^6$) |
| Exp 1 | 0.74 | 0.13 | 0.46 | 1.09 | 0.88 | 0.21 | 0.48 | 5.79 |
| Exp 2 | 0.85 | 0.39 | 0.55 | 1.18 | 0.95 | 0.66 | 0.70 | 13.91 |
| Exp 3 | 0.84 | 0.50 | 0.62 | 1.17 | 0.95 | 0.76 | 0.72 | 14.37 |

### 3.2.2   Estimation of water surface elevation

Although we used river discharge to evaluate assimilation efficiency, WSE is an important water dynamic estimator, in

particular for predicting floods. Table 4 summarizes the evaluation results of the assimilated using satellite altimetry measurements. We evaluated the $RMSE$, $BIAS$, and $\Delta A$ of WSE between assimilated and observed values. $RMSE$ represents the total departure from observations, whereas $BIAS$ denotes the difference in long-term mean values between simulation results and observations. The mean difference in the variation in the yearly peak and trough of the hydrograph was identified with $\Delta A$. Transitioning from direct to normalized value assimilation did not reduce $RMSE$ or $BIAS$. Nevertheless, the anomaly

and normalized assimilation methods improved the amplitude of WSE more than direct DA. Similar patterns were observed for the assimilation and validation VSs. The $BIAS$ of WSE, which accounts for a considerable portion of $RMSE$, was not corrected in the anomaly or normalized value assimilations.

Figure 9 illustrates the spatial distributions of the $RMSE$, $BIAS$, and $\Delta A$ of WSE for Exp 1 (Figure 9a), Exp 2 (Figure 9b), and Exp 3 (Figure 9c). Median $RMSE$ and $BIAS$ were lowest with direct DA (Exp 1), but $\Delta A$ was larger than in the anomaly and

normalized value DA experiments for all, assimilation, and validation VSs. $RMSE$ and $BIAS$ were lower along the lower Amazon mainstem and Negro River ($RMSE < 3m\ and\ BIAS < 2m$) compared to other river reaches. However, $\Delta A$ ($> 4m$) was less accurately estimated with direct DA than with other methods.

Large $RMSE$ values ($> 4m$) were obtained for the Madeira, upper Purus, and upper Solimoes rivers in anomaly DA (Exp 2; Figure 9b). Large $BIAS$ values occurred in the Amazon mainstem, Purus River, and Japura River, with $BIAS > 4m$. The




**Figure 9: Performance of water surface elevation estimation in terms of the root mean square error ($RMSE$), bias between assimilation results and observations ($BIAS$), and mean differences in amplitude between assimilation results and observations ($\Delta A$) for a) Exp 1, b) Exp 2, and c) Exp 3.**

annual variation in WSE (amplitude) differed considerably in some areas of the Amazon and Negro River mainstems ($\Delta A > 8m$).

With normalized DA (Exp 3), a large $RMSE$ ($> 4m$) was observed in the Madeira River, downstream reaches of the Amazon mainstem, and upper Purus River. Large $BIAS$ values occurred in the mid-section of the Amazon mainstem and Japura River ($BIAS > 6m$). $\Delta A$ was particularly high in some sections of the Amazon mainstem and Negro River ($|\Delta A| > 8m$). In

summary, direct DA estimated WSE with low $RMSE$ and $BIAS$ values, whereas the best $\Delta A$ was obtained with anomaly DA.

**Table 4: Median performance metrics for water surface elevation estimates obtained from DA experiments. Median values for the root mean square error ($RMSE$), long-term bias ($BIAS$), and difference in amplitude ($\Delta A$) are presented for all, assimilation, and validation VSs.**



| Experiment | All | | | Assimilation | | | Validation | | |
|---|---|---|---|---|---|---|---|---|---|
| | *RMSE* | *BIAS* | Δ*A* | *RMSE* | *BIAS* | Δ*A* | *RMSE* | *BIAS* | Δ*A* |
| Exp 1 | 4.56 | 2.38 | 3.86 | 4.58 | 2.39 | 3.68 | 4.25 | 1.90 | 4.75 |
| Exp 2 | 4.80 | 2.82 | 1.60 | 4.79 | 2.83 | 1.53 | 4.88 | 2.69 | 2.08 |
| Exp 3 | 4.80 | 2.76 | 1.75 | 4.78 | 2.74 | 1.72 | 4.96 | 2.84 | 2.10 |

### 3.3 Comparison of Discharge Products

To investigate the accuracy of river discharge estimation using DA compared to state-of-the-art hydrodynamic modeling, we compared river discharge obtained from CaMa-Flood forced with bias-corrected variable infiltration capacity LSM (Liang et al., 1994) runoff (Lin et al., 2019) data (VIC BC), direct DA (Exp 1), anomaly DA (Exp 2), and normalized value DA (Exp 3). We used VIC BC runoff (Lin et al., 2019) to force discharge without DA, whereas ensemble mean discharge was examined for all DA experiments. VIC BC runoff is produced with sparse cumulative density function matching, and combining VIC BC runoff with the CaMa-Flood hydrodynamic model yields more accurate discharge estimates (Lin et al., 2019). A comparison of boxplots showing *NSE* for various discharge products is presented in Figure 10. The median *NSE* for discharge determined using CaMa-Flood standard settings (CaMa VIC BC) was 0.42. Normalized value assimilation (Exp 3) provided

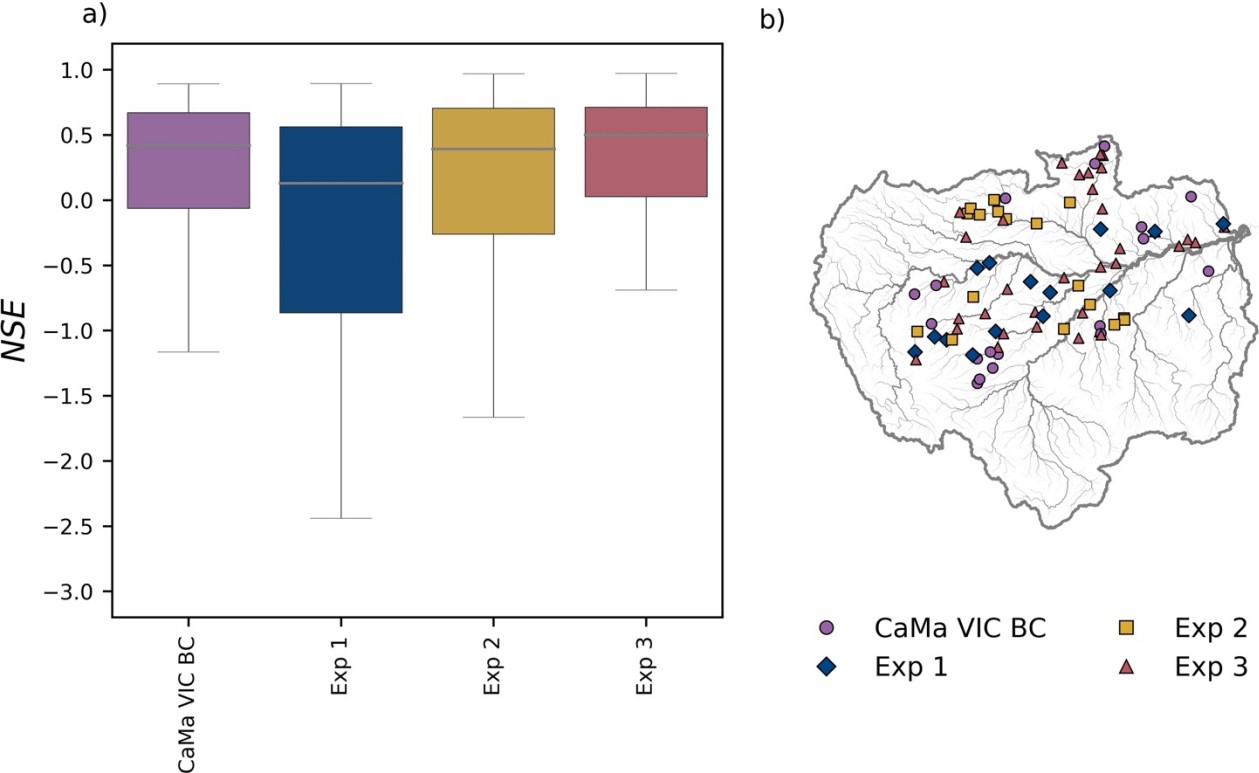

**Figure 10: a) Boxplot of Nash-Sutcliffe efficiency (*NSE*) for discharge simulated using CaMa-Flood with VIC bias-corrected runoff (CaMa-Flood VIC BC), mean assimilated with direct DA (Exp 1), mean assimilated with anomaly DA (Exp 2), and mean assimilated with normalized value DA (Exp 3). b) Spatial distribution of the best discharge estimates among CaMa VIC BC, Exp 1, Exp 2, and Exp 3 based on *NSE*.**



the best river discharge estimates, with a median *NSE* of 0.50, whereas direct and anomaly DA produced medians of 0.13 and
0.39, respectively. For normalized value DA, *NSE* values of river discharge were confined to around 0.5, with many of the
gauges demonstrating *NSE* values greater than zero. Hence, normalized value assimilation improved the *NSE* of river
discharge compared to standard CaMa-Flood modeling and the other DA methods tested (i.e., anomaly and direct assimilation).
Assessing the spatial distribution of the optimal discharge product is critical to improving discharge estimates globally. Figure
10b shows the heterogeneity of the optimal discharge estimate among the four products. River discharge was compared based
on the *NSE* of each discharge product. Exp 3, which accounted for 44% of the gauges, most accurately estimated discharge
overall. Exp 2 estimated river discharge more accurately for 24% of GRDC gauges. Direct DA provided better estimates for
23% of gauges, whereas CaMa VIC BC estimated river discharge accurately for 8% of gauges. Note that the discharge
estimates of some of the gauges located on the Amazon mainstem were better without DA. Most gauges with the most accurate
discharge estimates of CaMa-Flood VIC BC were located outside of satellite-observed river reaches and were marginally
affected by DA.

In conclusion, normalized value DA (Exp 3) performed best for estimating river discharge, but uncalibrated CaMa-Flood
simulations without DA (i.e., CaMa VIC BC) were more accurate than other DA methods (i.e., direct and anomaly
assimilation). Hence, assimilating satellite altimetry can improve the accuracy of river discharge estimates compared to current
state-of-the-art hydrodynamic modeling.

## 4  Discussion

### 4.1  DA Performance With Current Hydrodynamic Models

We compared several DA methods to overcome the errors associated with assimilating satellite altimetry observations directly
into a hydrodynamic model. Through the assimilation of anomalies or normalized values, river discharge estimation was
improved considerably compared to direct DA (Figure 7). Although WSE was correctly assimilated with direct DA (Figure
9a), river discharge estimates were inaccurate because of parameter errors and discrepancies in flow dynamics driven by
limited representation of actual physical phenomena (as illustrated in Figure 2). These biases can be caused by discrepancies
in parameters such as riverbank-full height and river bottom elevations. For example, when river channel depth was
overestimated in the model, simulated WSE was lower than the observations. When assimilated WSE was converted into the
CaMa-Flood prognostic variable (i.e., storage), the initial condition could be erroneous. Such errors were propagated to river
discharge. In areas where the simulated water dynamics (e.g., amplitude and flow regime) were similar to observations,
anomaly assimilation limited the extent to which the biases affected assimilation (Emery et al., 2020a; Paiva et al., 2013a;
Wongchuig-Correa et al., 2020). Although spurious errors (due to limited ensemble size) were regulated using physically-
based empirical localization patches (Revel et al., 2019), direct DA was adulterated because of the biases and dynamic
differences in WSE simulations.

Normalized value assimilation showed better assimilation efficiency in terms of *NSEAI*, representing the overall accuracy of
discharge estimation, compared to the anomaly DA method (Figure 7). Currently the CaMa-Flood hydrodynamic model cannot
accurately represent the dynamics of WSE because of limitations in the model framework (e.g., a lack of representation of
water regulations, diversions, and lake dynamics) and the impacts of water dynamics other than river flow. These limitations
also exist for most global hydrodynamic models, which do not accurately represent reservoirs, diversions, and lakes
(Fleischmann et al., 2021). For example, when reservoir operations are not represented in a hydrodynamic model, assimilating
observations obtained during reservoir operation will alter the flow regime. In addition, errors related to the model structure
hamper the prediction of surface water dynamics and may produce a different flow dynamic than the observations. Therefore,
normalized value assimilation provided the best estimates of river discharge given the current limitations of models such as
biases and poor representation of flow dynamics.

Moreover, the assimilation framework is computationally efficient and effective at removing spurious correlations. LETKF is
a computationally efficient filtering method that uses a local area for assimilation (Hunt et al., 2007; Miyoshi and Yamane,
2007). In addition, we used a physically-based empirical localization technique (Revel et al., 2019) to reduce erroneous
correlations and assimilate observations in significantly correlated areas. Hence, the assimilation framework is capable of



estimating river discharge at the global scale provided satellite observations are available. Once the SWOT satellite is launched,
the methods developed in this study will be valuable for accurately estimating river discharge.

## 4.2 DA Performance Under Various Conditions

To examine DA performance under model conditions such as biased runoff forcing and corrupted river bathymetry, we
performed biased runoff and corrupted bathymetry experiments. An artificial bias of –50% was introduced to the ensemble
mean of the "Runoff Ensemble" (Figure 1a) for each assimilation approach, namely, direct DA, anomaly DA, and normalized
value DA (Supplementary Text S1, Figure S3). Because of the bias introduced by the runoff forcing, river discharge and WSE
were approximately 50% lower in the open-loop simulation than in the observations. We artificially corrupted the river
bathymetry to represent errors in the hydrodynamic model (Supplementary Text S2, Figure S4). River channel depth was
increased by 25% in the corrupted bathymetry experiment. Then we assimilated satellite altimetry into the hydrodynamic
model with corrupted river bathymetry through the direct, anomaly, and normalized value DA methods. In general, the WSE
was reduced by approximately 25% of the river channel depth. For simplicity, we used only a single runoff (HTEESSEL;
Balsamo et al., 2011) from E2O WRR2 to prepare the runoff ensemble. The HTEESSEL runoff from E2O WRR2 is fairly

**Figure 11: Comparison of Nash-Sutcliffe efficiency ($NSE$) of assimilated river discharge under various conditions: a) without runoff bias or bathymetry error, b) without runoff bias and with bathymetry error, c) with runoff bias and without bathymetry error, and d) with runoff bias and bathymetry error. The direct, anomaly, and normalized value DA results are represented in blue, yellow, and red, respectively.**





unbiased (Dutra et al., 2017; Revel et al., 2021), and the default bathymetry parameter of CaMa-Flood should provide adequate WSE estimates (Yamazaki et al., 2012). Simulations using the default CaMa-Flood bathymetry parameter and HTEESSEL
runoff are referred to as "normal conditions".

Runoff bias and bathymetry errors affect the accuracy of assimilated river discharge in different ways. When neither runoff nor bathymetry was erroneous, the normalized value DA method performed best (median $NSE = 0.83$) at estimating river discharge in terms of $NSE$ (Figure 11a). The best river discharge estimates were obtained with anomaly DA (median $NSE = -0.45$) when the bathymetry contained some errors, but runoff was unbiased (Figure 11b). Bias in the runoff ensemble strongly
affected the accuracy of river discharge estimation with anomaly and normalized value DA, as bias in runoff causes bias in the mean and standard deviation used to generate WSE anomalies and normalized values (Figure 11c). Direct DA provided the best discharge estimation (median $NSE = 0.68$) when runoff was biased. When both runoff and river bathymetry were erroneous, none of the DA methods produced better discharge estimates than open-loop simulation. Therefore, the normalized DA method worked well under normal conditions, but anomaly DA produced better discharge estimates when the river
bathymetry had errors, and the direct DA method performed best under runoff-biased conditions. Simple calibration of the hydrodynamic model is recommended for successful normalized value DA (i.e., bias correction of runoff to obtain the mean discharge and river bathymetry calibration to accurately determine mean WSE).

### 4.3    DA Performance With Calibrated River Bathymetry

To investigate the performance of DA using a hydrodynamic model with calibrated river bathymetry, we used rating curve
calibration (Zhou et al., 2022) to correct the river bathymetry (Supplementary Text S3). Investigating the performance of DA with corrected bathymetry is essential, as river bathymetry is the most influential parameter for WSE (Brêda et al., 2019). Calibrating the river bathymetry increases the accuracy of the hydraulic relationship between discharge and WSE (i.e., the rating curve; (Zhou et al., 2022), thereby improving discharge estimation with direct DA (median $NSEAI = -0.50$; Figure 12). Minimization of the WSE bias attributable to river bathymetry improved discharge estimates obtained with the direct DA
method, although the anomaly and normalized value DA approaches had little effect on the estimation of river discharge (Figure 12b and c). River discharge estimation can be improved by updating river-related parameters. However, anomaly and normalized value assimilation (with and without river bathymetry calibration) had greater assimilation efficiencies than direct DA with the calibrated model. Therefore, correcting river-related parameters is essential to achieving good river discharge estimates with direct DA.
Furthermore, the bathymetry parameters calibrated with the rating curve approach were the most accurate values attainable under current conditions (Zhou et al., 2022). However, direct DA was not capable of producing more accurate river discharge estimates than other DA approaches (i.e., anomaly and normalized value DA; Figure 12). This finding indicates that calibrating a single parameter (i.e., river bathymetry) may be insufficient to improve the overall accuracy of river discharge estimation using direct DA.  Hence, calibrating other river-related parameters (e.g., riverbank height, floodplain profile, and cross-
sectional shape) is necessary to increase assimilation efficiency (median $NSEAI > 0$) when assimilating satellite altimetry data directly into large-scale hydrodynamic models such as CaMa-Flood.

Direct DA offers several benefits over anomaly or normalized value assimilation. Although the direct DA approaches reduced overall accuracy, the sharpness of the ensemble spread was substantially reduced compared to the anomaly and normalized value DA approaches (e.g., Figure 4c–e). In addition, the improvement in the accuracy of river discharge estimates with
anomaly or normalized value assimilation was lower in river reaches with high biases in open-loop runoff estimation (Figure S5). This finding suggests that direct DA methods can be used to correct river discharge values in river reaches where runoff causes large biases, but the river bathymetry parameter is reasonably accurate. By contrast, the reliability of discharge estimates in the anomaly and normalized DA experiments was highly dependent on the quality of the runoff ensemble. Therefore, direct assimilation has several advantages, such as greater confidence in DA-estimated river discharge and accurate discharge
estimation even when the runoff ensemble is biased.





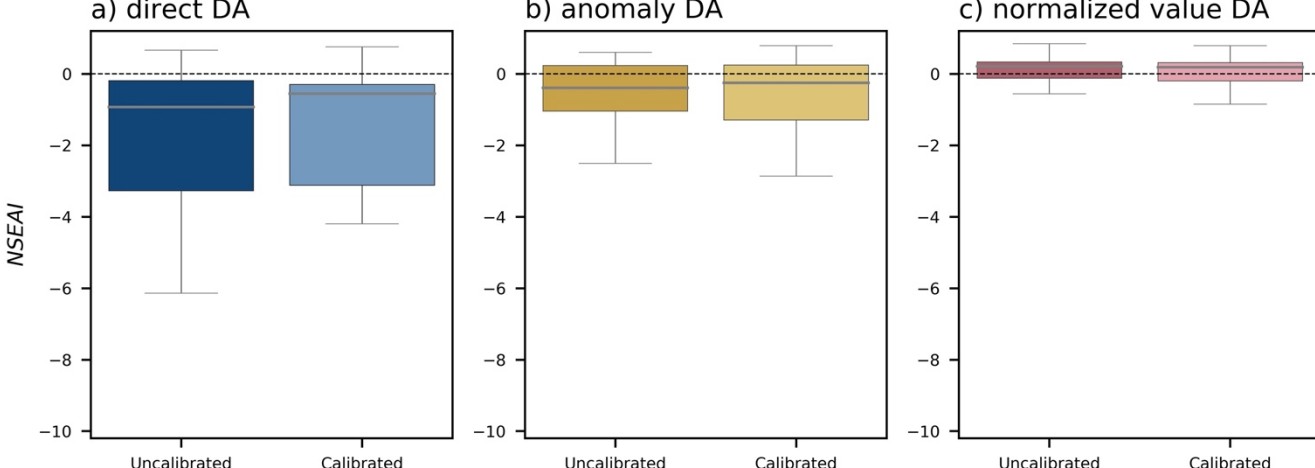

**Figure 12: Boxplot comparison of Nash-Sutcliffe efficiency-based assimilation index ($NSEAI$) values for uncalibrated and calibrated models with a) direct DA, b) anomaly DA, and c) normalized value DA.**

### 4.4    Potential and Limitations of River Hydrodynamics DA

The development of a discharge reanalysis product (such as those described by Feng et al., 2021; Wongchuig et al., 2019) is crucial to evaluating the reliability of assimilated discharge product within the capabilities of current hydrodynamic modeling. In addition, reanalyzes of river discharge play an important role in biodiversity and biogeochemistry research (Messager et al.,
2021). Discharge estimated from the assimilation of satellite altimetry characterized the flow dynamics of the Amazon Basin better than estimates from a state-of-the-art hydrodynamic model (Figure 10). However, CaMa-Flood modeled river discharge better than the assimilated product in certain river reaches along the Amazon mainstem. These discrepancies are primarily due to the limitations of hydrodynamic modeling, as the assimilated WSEs were adequately represented in the assimilated simulation. As we assimilated WSE and corrected the initial conditions of the following time step using CaMa-Flood
parameters (e.g., riverbank height, river bathymetry, river width, and floodplain profile), the errors of the modeling framework may have propagated into the river discharge estimates at the next time step. These limitations can be circumvented through the assimilation of in situ or remotely sensed river discharge observations into hydrodynamic models (Emery et al., 2020b; Feng et al., 2021; Paiva et al., 2013a; Wongchuig et al., 2019). Yet with decreasing numbers of in situ gauges (Hannah et al., 2011; Shiklomanov et al., 2002; Vörösmarty et al., 2001) and low accuracy of remotely sensed river discharge estimates
(Bjerklie et al., 2018; Gleason and Durand, 2020; Gleason and Smith, 2014), obtaining consistent and reliable observations can be difficult.
DA of satellite altimetry had several advantages over hydrodynamic modeling, in particular when it came to accurately estimating low flows and unanticipated peaks that were not reflected in the runoff forcing (Figure S6). These unexpected peaks, which were not as large as annual peaks, were characterized well by DA methods (Figure S6), although the open-loop
simulation did not identify them. Low flows were estimated well with normalized assimilation and further improved through calibration of river bottom elevation (Supplementary Text S4, Figure S7). Hence, the DA scheme accurately represented low flows and unforeseen secondary peaks.
The normalized DA approach may have been unable to accurately predict other variables, such as WSE and flood extent, as the assimilation was performed in transformed space. WSE estimation using normalized value DA had lower overall accuracy
than direct DA (Figure 7) based on median $RMSE$ (Table 4). Moreover, flood extent would be better estimated with direct DA than other DA methods, as flood extent is diagnosed with WSE in the CaMa-Flood hydrodynamic model. Hence, normalized DA may be unable to effectively predict various important variables (e.g., WSE and flood extent).





## 5    Conclusion

In this study, we explored strategies for assimilating satellite altimetry data into a contemporary hydrodynamic model. As
existing large-scale hydrodynamic models either are too conceptual or have uncertainties in their parameter schemes, direct or
anomaly assimilation of satellite altimetry may introduce inaccuracies due to discrepancies between satellite altimetry and
simulated WSE. We assessed direct, anomaly, and normalized value DA schemes using a continental-scale hydrodynamic
model, CaMa-Flood (Yamazaki et al., 2011). We used the physically based localization approach of LETKF to assimilate
satellite altimetry data in the Amazon Basin. Normalized value assimilation performed better than other approaches at
estimating river discharge in this continental-scale river basin. River discharge was accurately estimated with normalized value
assimilation in most river reaches covered by satellite observations ($NSE > 0.6$).

We investigated the capacity of DA approaches to reliably estimate river discharge through cutting-edge hydrodynamic
modeling. River discharge was well characterized in the normalized value assimilation experiments, with a median $NSE \approx$
0.47 , which was better than the river discharge produced by the uncalibrated model with default parameters using HTEESSEL
runoff (Balsamo et al., 2011) (median $NSE \approx 0.13$). The median $NSE$ of river discharge improved by 34% with the
assimilation of satellite altimetry into a continental-scale hydrodynamic model. Improvements were evident across the entire
Amazon Basin; however, some degradation occurred due to underestimation of peak river discharge in the Amazon mainstem.
This underestimation of peaks may be attributable to uncertainties in other parameters of the hydrodynamic model.

The estimation of river discharge using DA methods is variable and depends on the state of the runoff data (i.e., bias) and the
accuracy of river cross-section parameters (i.e., river bathymetry). When the runoff was biased and lacked river bathymetry
error, the direct DA approach performed best. When the river bathymetry was erroneous, anomaly DA performed best.
However, when runoff was biased and river bathymetry was erroneous, none of the DA methods performed better than open-
loop simulation. Hence, depending on the runoff and river bathymetry, different DA approaches should be used. To realize the
advantages of the normalized value DA approach, basic model calibration is necessary, such as calibration of runoff to capture
the mean discharge and moderate calibration of bathymetry to capture WSE patterns.

River bathymetry calibration enhanced the accuracy of the river discharge estimates produced using the direct DA method but
had minimal effect on normalized assimilation. Zhou et al., (2022) used a calibration strategy to increase the accuracy of river
bathymetry by decreasing WSE error utilizing the stage-discharge relationship; they found that the approach does not
necessarily improve river discharge accuracy. In addition, when the calibrated model was forced by runoff with large errors,
normalized DA did not improve the estimation of river discharge because of bias in the mean discharge and WSE. The quality
of runoff perturbation data should be evaluated before they are used in anomaly or normalized value assimilations.

The use of precise river cross-section estimates, and floodplain dynamic processes may improve peak discharge estimates. To
represent the river discharge more accurately, some improvements to the CaMa-Flood hydrodynamic model may be necessary.
Furthermore, assimilating multiple variables such as river discharge, WSE, and flooded area may improve discharge estimates
further. Overall, the methods developed in this study demonstrate great potential for using available satellite altimetry to
improve river discharge estimation in continental-scale rivers within the limitations of current hydrodynamic models.

## 6    Code Availability

The DA code is open source and freely available from https://github.com/MenakaRevel/HydroDA.git. CaMa-Flood is freely
available    from    http://hydro.iis.u-tokyo.ac.jp/~yamadai/cama-flood/    or    git@github.com:global-hydrodynamics/CaMa-
Flood_v4.git (Yamazaki et al., 2011).

## 7    Data Availability

The key data sets used in this study are available from https://doi.org/10.4211/hs.08e1b18aa9f240758dd13d9ac875621f (Revel
et al., 2022). The source code used for data assimilation (DOI: 10.5281/zenodo.6506861) is publicly available. Satellite





altimetry data used in this study can be obtained from https://hydroweb.theia-land.fr/. GRDC river discharge observations are
available from https://portal.grdc.bafg.de. Runoff data of E2O WRR2 can be accessed at https://wci.earth2observe.eu.

## 8   Author Contributions

MR and DY designed the experiments. MR developed the code for the simulations. XZ provided data for the experiments.
MR, XZ, DY, and SK reviewed and edited the manuscript.

## 9   Competing Interests: The authors have no conflicts of interest to declare.

## 10   Acknowledgment

This work was supported by the Japan Society for the Promotion of Science (JSPS) under KIBAN-S Grant No. 21H05002 and
JSPS KIBAN-B 20H02251.

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
