# Peer review of "Assimilation of Transformed Water Surface Elevation to Improve River Discharge Estimation in a Continental-Scale River"

_EGUsphere, 2022_

## Author Response (AR1)

**Editor decision: Publish subject to revisions (further review by editor and referees)**

Comments to the author:
Dear Authors

Three reviewers looked at the manuscript all are positive about the scientific quality. However they provide good suggestions improve the manuscript further. P

Sincerely,

Albrecht Weerts

Reply: We would like to express our gratitude to the editor and the anonymous referees for their thorough examination of the manuscript. The comments and suggestions greatly improve the manuscript. The point-to-point answers are presented below for each referee comment. The answers to the referee's comments were written in blue. The changes which were made to the text of the manuscript were presented in *italics* in quotation (" … ") marks.

**Referee #1**

General comment:

In this research, the authors performed data assimilation (DA) experiments to explore the capacity to improve daily river discharge within current limitations of global hydrodynamic modeling. For this purpose, the water surface elevation (WSE) from satellite altimetry was assimilated in a configuration of three experiments, the direct (absolute values), the anomalies and the normalized anomalies. The authors also evaluated the capability of these DA experiments in some scenarios, for instance when some parameters/forcing (river bathymetry and runoff) of the hydrodynamic model are biased, as well as conditions when river bathymetry is calibrated. The results showed that, in general, the normalized DA performance was the best, improving the daily discharge estimates in up to almost 60% of the stations evaluated, compared to the simulation without DA. These results considering the current limited conditions of the global hydrodynamic models (e.g. without calibration).

The major contribution of this research is the evaluation of these experiments and scenarios, providing adequate knowledge and insights in terms of how DA techniques could be used to improve discharge estimates, which fits with the perspectives of SWOT missions for example. In general, this work is worth publishing in the Hydrology and Earth System Sciences journal, however, it needs "moderate revisions". Some suggestions for revisions are as follows.

Reply:

We would like to thank the referee for his insightful comments. We addressed all the comments in the revised manuscript and detailed responses to the comments are given below.

Moderate comments:

1. The article is well written and follows a logical order. Although it is a bit long, an average reader can follow the reading, however at a certain point there are more experiments than

initially described. For example, the evaluation of DA under biased runoff and river bathymetry conditions; DA under calibrated river bathymetry conditions; DA using the runoff forcing of a bias-corrected model. That is why I recommend the authors to describe more explicitly these experiments in section 2.6.

Reply:

Thank you very much for the recommendation. We included brief descriptions of the additional experiments in section 2.6. But we also think that the manuscript will become lengthy by adding more text. Therefore, we included the following in the main text and kept some explanations in the supplementary information.

*"Moreover, we conducted biased runoff (Supplementary Text S2, Figure S4) and corrupted river bathymetry (Supplementary Text S3, Figure S5) experiments to further understand DA performance under different model conditions.".*

2. Regarding the selection of virtual stations (VSs) for assimilation or validation, the justification is a bit vague, even though this may be important for the performance of the experiments, so I recommend improving this point.

Reply:

We would like to express our gratitude to the referee. We simply separated VSs into assimilation and validation as 80%, and 20%, respectively. We revised the description as follows.

*"Furthermore, we randomly chose 80% of the VSs in the Amazon basin for assimilation, while the remaining 20% were preserved for WSE validation (Figure 3). Consequently, we used only 80% of the 324 VSs in the Amazon basin for the DA."*

3. In sections 3.1.1, 3.1.2 and 3.1.3 take care with the description of the time series in figures 4, 5 and 6, respectively. There is a confusion between the description of the gauges results. For example, line 315 describes the Santos Dumont station on the Purus river, however the series in Figure 4d are from a gauge on the Juruá river. This confusion occurs for the stations Gaviao (Juruá) and Manacapurú (Amazon) in Figure 5, and in all the gauges in Figure 6. This was probably an involuntary error in the preparation of the figures, please correct.

Reply:

We thank referee #1 for recognizing the mistake. It was indeed an involuntary error in preparing the figures. We corrected the cross-reference with the corresponding figure and the text. Furthermore, we corrected the manuscript according to the referee's comment. Basically, we replaced Figures 4, 5, and 6 to match the descriptions in Sect. 3.1.1, 3.1.2, and 3.1.3, respectively.

The revised Figures 4, 5, and 6 were as follows:

[Figure]

*Figure 4: a) Difference in the correlation coefficient of river discharge (Δr) and b) relative Root Mean Square Error (rRMSE) of water surface elevation for DIR. Circles indicate virtual stations used for data assimilation, and squares are virtual stations used for validation on the WSE plots. Hydrographs recorded at Labera on the Purus River, Santos Dumont on the Jurua River, and Santo Antonio Do Ica on the Amazon River are presented in panels c, d, and e, respectively. The locations of the hydrographs shown in panels c, d, and e are presented in panel a. Discharge observations are shown in black, assimilated simulation results in orange, and open-loop simulation results in blue. The color range indicates the 95% confidence interval used to calculate the relative interval skill score (rISS). Δr, Nash-Sutcliffe efficiency-based assimilation index (NSEAI), and rISS are shown at the*

[Figure]

*Figure 5: a) Difference in the correlation coefficient of river discharge (Δr) and b) relative Root Mean Square Error (rRMSE) of water surface elevation for DIR. Circles indicate virtual stations used for data assimilation, and squares are virtual stations used for validation on the WSE plots. Hydrographs recorded at Gaviao on the Jurua River, Manacapuru on the Amazon River, and Serrinha on the Negro River are presented on panels c, d, and e, respectively. The locations of the hydrographs shown in panels c, d, and e are presented in panel a. Discharge observations are shown in black, assimilated simulation results in orange, and open-loop simulation results in blue. The color range indicates the 95% confidence interval used to calculate the relative interval skill score (rISS). Δr, Nash-Sutcliffe efficiency-based assimilation index (NSEAI), and rISS are shown at the bottom of each panel.*

[Figure]

*Figure 6: a) Difference in the correlation coefficient of river discharge (Δr) and b) relative Root Mean Square Error (rRMSE) of water surface elevation for DIR. Circles indicate virtual stations used for data assimilation, and squares are virtual stations used for validation on the WSE plots. Hydrographs recorded at Sao Paulo De Olivenca on the Amazon River, Vila Bittencourt on the Japura River, and Curicuriari on the Negro River is presented on panels c, d, and e, respectively. The locations of the hydrographs shown in panels c, d, and e are presented in panel a. Discharge observations are shown in black, assimilated simulation results in orange, and open-loop simulation results in blue. The color range indicates the 95% confidence interval used to calculate the relative interval skill score (rISS). Δr, Nash-Sutcliffe efficiency-based assimilation index (NSEAI), and rISS are shown at the bottom of each panel.*

4. Experiments that assimilate absolute (direct) values, anomalies and normalized anomalies are referred to by the acronyms Exp. 1, Exp. 2 and Exp. 3 respectively, however throughout the manuscript both nomenclatures are used. I suggest that only one be adopted to improve the readability of the text. Even so that the information can be quickly abstracted by the reader these experiments could be called DIR_DA, ANOM_DA and NORM_DA for example.

Reply:

We appreciate the suggestion from referee #1. We adopted a new naming convention for the experiments in the revised manuscript as DIR, ANO, and NOM for the direct, anomaly, and normalized values DA, respectively instead of Exp 1, Exp 2, and Exp 3.

5.  Since the authors have used a localization method in the DA scheme, I suggest reinforcing the discussion on how this might affect discharge estimates due to assimilation of WSE within or outside the influence coverage of the VSs.

Reply:

We would like to thank the great suggestion by referee #1. The localization method we used in this study is an adaptive localization method (Revel et al., 2019) which is far different from conventional localization methods which used fixed square-shaped local patches. The comparison between those methods can be found in Revel et al., (2019). The adaptive localization method recognizes the highly correlated areas and removes less correlated areas (e.g., small downstream tributaries). We highlighted the effect of adaptive localization in the revised manuscript as follows.

*"Moreover, the assimilation framework is computationally efficient and effective at removing spurious correlations. LETKF is a computationally efficient filtering method that uses a local area for assimilation (Hunt et al., 2007; Miyoshi and Yamane, 2007). In addition, we used a physically-based empirical localization technique (Revel et al., 2019) to reduce erroneous correlations and assimilate observations in significantly correlated areas. It has been found that the physically-based empirical localization method performed better than the conventional square-shaped local patches in hydrodynamic DA schemes (El Gharamti et al., 2021; Ishitsuka et al., 2020; Revel et al., 2019; Wongchuig et al., 2019). Hence, the assimilation framework is capable of estimating river discharge at the global scale provided satellite observations are available. Once the SWOT satellite is launched, the methods developed in this study will be valuable for accurately estimating river discharge."*

6.  Could you discuss a bit about to what do you attribute the lower efficiency of flow estimates in the upper Solimoes River ? efficiency of the CaMa-Flood model ? selection of VSs ? localization ? large uncertainties in the VSs data in that region ?

Reply:

We would like to thank referee #1 for raising the question. We presume that the referee is referring to the lower assimilation efficiency of the normalized value DA method compared to the direct DA method in the upper Solimoes River. In the upper Solimoes River, we compared two gauge locations namely, Santo Antonio Do Ica and Sao Paulo De Olivenca. These two locations show slightly higher NSE values than the anomaly and normalized value DA methods. Mostly the reason has been the underestimation of the peak discharge. The underestimation of peak discharge may be due to the underestimation of runoff. We attached Figure R1 for a comparison of river discharge for Santo Antonio Do Ica and Sao Paulo De Olivenca. Table R1 indicates the NSE values for the upper Solimoes Rivers.

Table R1: NSE of Santo Antonio Do Ica and Sao Paulo De Olivenca

|  | Santo Antonio Do Ica | Sao Paulo De Olivenca |
| --- | --- | --- |
| Open loop | -1.05 | -0.61 |
| DIR | -1.00 | -0.42 |
| ANO | -1.07 | -0.59 |
| NOM | -1.08 | -0.57 |

[Figure]

Figure R1: Example hydrographs of Upper Solimoes River. Santo Antonio Do Ica and Sao Paulo De Olivenca gauging location are presented. Observations, Open-loop, Direct DA (DIR), Anomaly DA (ANO), and Normalized value DA (NOM) are presented in black, grey, blue, yellow, and red, respectively.

Specific comments (Line-by-line comments):

Introduction:

1.  35: It is more accurate to say, "River discharge records can be used...".

    Reply: We revised the sentence as follows.

    *"River discharge records can be used to assess water resources, biogeochemistry, and the carbon cycle in terrestrial waters and is the single most important parameter affecting the flow dynamics of the rivers (Gleason and Durand, 2020)."*

2.  42: I would say that also these simulations (of GHMs) have been used to complement observed records.

    Reply: We agree with referee #1 that the simulations of GHSs have been used rather complement observations rather than compensate. We revised the text as below.

    *"Simulated water dynamics obtained from GHMs are used to complement unavailable ground observations."*

3.  48: If we go deeper we could say that these forcing factors can also be rainfall and climatic variables.

    Reply: We modified the text to the following form.

    *"Uncertainties in the forcing are also partially responsible for uncertainty in the surface water dynamics (Emery et al., 2020c)."*

4. 50: I would say: "Given the current limitations of GHMs and in-situ measurements, …".

   Reply: The text was revised as shown below.

   *"Given the current limitations of GHMs and in situ measurements, satellite altimetry observations provide an alternative method of estimating the surface water dynamics (Feng et al., 2021)."*

5. 57: This sentence mentioning the SWOT mission seems a bit loose, you should rework it to integrate it with what you want to mention above.

   Reply: The sentence was revised as given below.
   *"The Surface Water and Ocean Topography (SWOT) satellite will provide an unprecedented amount of data for the first time on terrestrial waters (Biancamaria et al., 2016; Fu et al., 2012)."*.

6. 62: Typo: "combining" instead of "combing".

   Reply: We corrected it as shown below.

   *"Surface water dynamics can be clarified by combining remote sensing data with a limited amount of observational data in continental-scale hydrodynamic models."*.

7. 83: This statement describes information repeated in the previous one, perhaps you could combine them.

   Reply: The paragraph was modified as follows.

   *"Although DA approaches can improve model performance, hydrodynamic models are not yet mature enough to directly assimilate satellite altimetry data (Emery et al., 2020a). Because of ambiguity in digital elevation models (DEMs), flaws in hydraulic parameters (e.g., river bathymetry), and the simplification of cross-section parameters, simulated WSE may have substantial errors. Several methods have been used to circumvent these limitations when assimilating satellite altimetry into large-scale hydrodynamic models, including assimilating anomalies (i.e., removing the long-term mean WSE) and using a common datum (e.g. Emery et al., 2020a; Michailovsky et al., 2013; Paiva et al., 2013a; Wongchuig-Correa et al., 2020). To improve river discharge estimation in the Brahmaputra River, Michailovsky et al., (2013) assimilated measurements from the ENVISAT satellite into a rainfall-runoff model using a common reference for satellite altimetry and simulated river depth (i.e., adding the difference between modeled river depth and altimetry elevation to satellite altimetry). Likewise, anomalies from ENVISAT observations were assimilated into a continental-scale hydrologic/hydrodynamic model and compared to in situ and remotely sensed river discharge data in the Amazon Basin (Paiva et al., 2013a). Moreover, global-scale hydrodynamic modeling studies have used anomaly assimilation to eliminate biases in simulated WSE (Brêda et al., 2019; Emery et al., 2020a; Paiva et al., 2013a; Wongchuig-Correa et al., 2020). However, anomaly assimilation does not provide accurate river discharge estimates for the Amazon Basin (Paiva et al., 2013a), as it cannot compensate for discrepancies in flow dynamics between observations and simulations. These differences in flow dynamics can be attributed to*

*several factors, including differences in amplitude due to limited river width (De Paiva et al., 2013; Yamazaki et al., 2012), differences in seasonal flow due to failure to capture anthropogenic activity (Hanazaki et al., 2022; Pokhrel et al., 2018; Shin et al., 2020), and differences in flow variation due to the assumption of rectangular cross-sections (Neal et al., 2015; Saleh et al., 2013). Given such uncertainties in parameters and the structural simplification of current hydrodynamic models, anomaly assimilation of satellite altimetry may not be effective for estimating river discharge (Liu et al., 2012; Paiva et al., 2013a). Therefore, alternative approaches to direct and anomaly assimilation are required to integrate satellite altimetry into existing hydrodynamic models."*

Methodology:

1. 124: In this sentence you can already start reporting on the period of DA experiments (2009-2014).

   Reply: We would like to thank Referee #1 for the nice suggestion. But we try to dedicate this section (2.1) solely to describing the assimilation framework. So, we introduced the period of the experiments in section 2.6.

2. 215: Why wasn't the SURFEX-TRIP model outputs used since it also belongs to WRR2?

   Reply: We would like to thank the referee for the question. We did not consider outputs from the SURFEX-TRIP model because those are not compatible with the CaMa-Flood model. The SURFEX-TRIP model consists of the capillary rise in the runoff variable where CaMa-Flood is not capable of dealing with such runoff data. Therefore, we added the following sentences to the paragraph.

   *"The SURFEX-TRIP (Vergnes et al., 2014) model outputs were not used since they were incompatible with the CaMa-Flood hydrodynamic model."*

3. 236: To reference these annual average rainfall values you can cite Builes-Jaramillo & Poveda, 2018; Espinoza et al., 2009. (https://doi.org/10.1029/2017WR021338 and https://doi.org/10.1002/joc.1791)

   Reply: We would like to express our gratitude to referee #1. We cited the above studies when describing the annual average rainfall in the Amazon basin and the text was modified as follows.

   *"This basin receives substantial annual rainfall (≈2200 mm) with high spatial heterogeneity and experiences distinct rainy and dry seasons in the southern and eastern portions (Builes-Jaramillo and Poveda, 2018; Espinoza Villar et al., 2009)."*

4. 237: Please specify what you mean by large number of observations, perhaps this is valid for remote sensing observations because it is a large basin with strong hydrological signals, hence the citation of Fassoni-Andrade et al. 2021.

   Reply: We thank referee #1 for raising this issue. Yes, indeed we refer to remote sensing data in the Amazon basin. So, we revised the text as the following sentence.

*"The major advantage of analyzing the Amazon Basin is the availability of a large number of remote sensing observations (Fassoni-Andrade et al., 2021)."*

5. 249: You could elaborate a little more on this sentence. Why these virtual stations could affect the estimates using assimilation? this exclusion of 3% was by a visual analysis of the series only? these stations are located in some particular place in the Amazon, maybe rivers with a small width?

   Reply: We would like to express our gratitude to referee #1 for the important question. We did some extensive analysis of these largely biased VS which cannot be compared with the MERIT DEM (Yamazaki et al., 2017). This exclusion was done by comparing the mean WSE of satellite altimetry observation with the MERIT DEM, not on visual analysis. Our analysis revealed that most of the "biased VS" were in narrow rivers at relatively high elevations. We included the following in the revised manuscript. We are indeed preparing a manuscript on how to compare satellite altimetry observation with the large-scale hydrodynamic model outputs.

   *"These VSs were in narrow rivers at relatively high elevations."*

6. Sections 2.7.1 and 2.7.2 could be merged, as it could confuse the reader. The main objective of this research is to evaluate the performance in simulating daily discharge but here also the performance of WSE will be evaluated. This merged section could be called "observational data" since the altimetry data has also been used for validation.

   Reply: We appreciate the comment on merging Sections 2.7.1 and 2.7.2. We merged them in the revised manuscript.

Results:

1. 294-296: This sentence seems to be repetitive with the previous one, you could merge them.

   Reply: We would like to thank referee #1 for the comment. We revised the text as follows.

   *"In this section, we present the relative performance of each assimilation approach, in order of, the direct (DIR), anomaly (ANO), and normalized value (NOM) DA experiments. Here we analyze the performance of assimilated values with respect to the open-loop simulation. Δr represents the relative change in r between the open-loop or assimilation results and observations. rRMSE represents the deviation in the RMSE of assimilation relative to that of open-loop simulation. rISS, rSharpness, and ΔReliability are used to assess changes in ensemble spread between the open-loop and assimilated simulations followed by a comparison of the relative performance (i.e., Δr, NSEAI, KEGAI, rISS, rSharpness, and ΔReliability) of among experiments (i.e., DIR, ANO, NOM)."*

2. 302: It would be appropriate to refer to Figure 4b in this sentence.

   Reply: Thank you very much for pointing out this. We revised the text to refer to figure 4d in the sentence.

*"The relative change in RMSE between observed and simulated WSE (rRMSE) was used to evaluate WSE performance, as illustrated in Figure 4b."*

3. 309: The time series for the Santos Dumont station is not shown in Figure 4d. Instead, a station on the Juruá River is shown. See my major comments above.

   Reply: Thank you very much for the comment. We revised Figure 4d to be compatible with the description in the text as shown in the answers' main comment #3.

4. 325: The information in parentheses should go in the methodology section.

   Reply: Thank you for the kind suggestion. We included the following in the methodology section.

   *"The statistics (i.e., mean and standard deviation) for satellite altimetry were determined from the period when observations were available. However, for WSE simulations, we obtained statistics using simulations from 2000 to 2014."*

5. 330: "WSE performance decreased…" instead of "WSE decreased…".

   Reply: Thank you very much. We revised it as below.

   *"WSE performance decreased (in terms of RMSE) in the Jurua and Purus rivers, although nearly all other river reaches showed increases in the accuracy of WSE calculations with DA."*

6. 332: The Gavião and Manacapuru gauges do not correspond to Figures 5c and d.

   Reply: Thank you. We revised figures 5c and d to match the description as shown in the answers to main comment #3.

7. 351-352: None of these described gauges correspond to figures 6c, d and e.

   Reply: Thank you for raising this error. We replaced the correct figures for figures 6c, d, and e; and the description was revised to correspond to figures 6c, d, and e.

8. 412: I think there is a typo, please delete "3.2.1.".

   Reply: Thank you for pointing out it. We deleted "3.2.1.".

9. 8: It is not possible to distinguish gauges inside or outside the coverage area of the altimetric satellites. Could you differentiate them somehow?

   Reply: Thank you for the great suggestion. We modified figure 8 to distinguish the gauges inside and outside satellite altimetry coverage. We attached the revised figure as follows.

10. Table 3. I have noticed that some values in this table do not correspond exactly to those described. For instance, in the first column (All and r) in the table, the values are 0.74, 0.85

and 0.84 for experiments 1, 2 and 3 respectively. While in the description the values are 0.73, 0.84 and 0.83 (L. 431, L. 415 and L. 439 respectively).

Reply: Thank you for cross-checking the description with the data provided in the tables. The values in the table were correct values so we revised the text according to Table 3. We revised the L. 431, L. 435, and L. 439 with the values of 0.74, 0.85, and 0.84. The revised text is as follows.

*"The r of river discharge estimation for several GRDC gauges was >0.8 (approximately 38%), with a median r of 0.74."*

*"The spatial distribution of the absolute performance of river discharge estimation through anomaly DA is shown in Figure 8b. Anomaly DA (ANO) produced r values of river discharge that were >0.8 in 84% of gauges, with a median r=0.85."*

*"Nearly 57% of GRDC gauges had r>0.8, with a median r of 0.84 in NOM."*

[Figure]

*Figure 8: Performance of daily discharge in terms of the correlation coefficient (r), Nash-Sutcliffe efficiency (NSE), and Kling-Gupta efficiency (KGE) for a) DIR, b) ANO, and c) NOM. Diamonds and circles indicate the gauges outside and inside the satellite altimetry coverage, respectively.*

11. 452: As shown in Figure 9, the BIAS values are only positive, so I recommend describing somewhere (probably methodology) that the index is an absolute value of BIAS.

Reply: Thank you for pointing out this. We added "absolute bias" to the methodology section when introducing the "BIAS" term as follows.

*"Furthermore, RMSE, the absolute bias between the means of observations and simulation results (BIAS),".*

12. Figure 9: It is a bit difficult to differentiate the VSs that were used for assimilation and validation. Perhaps it could be improved by changing the symbology from "o" to "*", increasing the size of the maps by reducing the space between them and decreasing a little the size of the station symbols so that they do not overlap too much. This is just a suggestion.

Reply: Thank you very much for your precious suggestion. We revised figure 9 and other similar figures to better represent symbols. We attached the revised figure 9 below.

[Figure]

*Figure 9: Performance of water surface elevation estimation in terms of the root mean square error (RMSE), bias between assimilation results and observations (BIAS), and mean differences in amplitude between assimilation results and observations (ΔA) for a) DIR, b) ANO, and c) NOM.*

13. Section 3.3: Please detail how in this experiment you have generated the realizations of the set for assimilation. Was it with the same perturbation as for the WRR2 models?

Reply: Thank you for asking for clarification. This section used the same assimilation outputs from the direct, anomaly, and normalized DA experiments. The perturbations were similar to WRR2 model outputs. On the other hand, we compared DA outputs with model simulation with the best runoff forcing (i.e., VIC BC: Lin et al., (2020)).

14. 476-478: The end of this sentence sounds strange, I suggest to redo it or delete this last part from "...,direct DA (Exp 1)...."

Reply: Thank you. We revised the sentence.

*"To investigate the accuracy of river discharge estimation using DA compared to state-of-the-art hydrodynamic modeling, we compared river discharge obtained from CaMa-Flood forced with bias-corrected variable infiltration capacity LSM (Liang et al., 1994) runoff data (VIC BC: Lin et al., 2019)."*

Conclusions:

1. 624: Typo, it's HTESSEL not HTEESSEL (same for L.541, 542 and 544).

Reply: Thank you, We corrected them to be HTESSEL in L541, 542, and 544.

*"For simplicity, we used only a single runoff (HTESSEL; Balsamo et al., 2011) from E2O WRR2 to prepare the runoff ensemble. The HTESSEL runoff from E2O WRR2 is fairly unbiased (Dutra et al., 2017; Revel et al., 2021), and the default bathymetry parameter of CaMa-Flood should provide adequate WSE estimates (Yamazaki et al., 2012). Simulations using the default CaMa-Flood bathymetry parameter and HTESSEL runoff are referred to as "normal conditions".*

Referee #2

The authors investigated the assimilation of satellite altimetry to improve discharge. The article is excellent. The authors presented 3 types of altimetry assimilation, which are 1. direct, 2. anomaly, and 3. normalized assimilation, and concluded that to improve discharge it is better to just assimilate the water surface dynamics (3.) given that the model is relatively accurate. If the model is completely corrupted, the authors concluded that anomaly and/or direct assimilation are more effective.

Reply:

We would like to express our gratitude to referee #2 for his thoughtful comments. We will address all the comments in the revised manuscript and detailed responses to the comments are given below.

1. My main complaint about this manuscript is in section 3.1 (specifically 3.1.1, 3.1.2, and 3.1.3) where the figures (fig 4, 5 and 6) showed hydrographs that were different from the locations discussed in the text. Besides that, I have just some small comments that are specified below:

   Reply:

   We would like to thank referee #2 for identifying this issue. We found that figures 4, 5, and 6 are wrongly attached. Therefore, we revised figures 4, 5, and 6 to be compatible with the description in sections 3.1.1, 3.1.2, and 3.1.3.

   The revised figures were attached below.

[Figure]

*Figure 4: a) Difference in the correlation coefficient of river discharge (Δr) and b) relative Root Mean Square Error (rRMSE) of water surface elevation for DIR. Circles indicate virtual stations used for data assimilation, and squares are virtual stations used for validation on the WSE plots. Hydrographs recorded at Labera on the Purus River, Santos Dumont on the Jurua River, and Santo Antonio Do Ica on the Amazon River are presented in panels c, d, and e, respectively. The locations of the hydrographs shown in panels c, d, and e are presented in panel a. Discharge observations are shown in black, assimilated simulation results in orange, and open-loop simulation results in blue. The color range indicates the 95% confidence interval used to calculate the relative interval skill score (rISS). Δr, Nash-Sutcliffe efficiency-based assimilation index (NSEAI), and rISS are shown at the bottom of each panel.*

[Figure]

*Figure 5: a) Difference in the correlation coefficient of river discharge (Δr) and b) relative Root Mean Square Error (rRMSE) of water surface elevation for DIR. Circles indicate virtual stations used for data assimilation, and squares are virtual stations used for validation on the WSE plots. Hydrographs recorded at Gaviao on the Jurua River, Manacapuru on the Amazon River, and Serrinha on the Negro River are presented on panels c, d, and e, respectively. The locations of the hydrographs shown in panels c, d, and e are presented in panel a. Discharge observations are shown in black, assimilated simulation results in orange, and open-loop simulation results in blue. The color range indicates the 95% confidence interval used to calculate the relative interval skill score (rISS). Δr, Nash-Sutcliffe efficiency-based assimilation index (NSEAI), and rISS are shown at the bottom of each panel.*

[Figure]

*Figure 6: a) Difference in the correlation coefficient of river discharge (Δr) and b) relative Root Mean Square Error (rRMSE) of water surface elevation for DIR. Circles indicate virtual stations used for data assimilation, and squares are virtual stations used for validation on the WSE plots. Hydrographs recorded at Sao Paulo De Olivenca on the Amazon River, Vila Bittencourt on the Japura River, and Curicuriari on the Negro River is presented on panels c, d, and e, respectively. The locations of the hydrographs shown in panels c, d, and e are presented in panel a. Discharge observations are shown in black, assimilated simulation results in orange, and open-loop simulation results in blue. The color range indicates the 95% confidence interval used to calculate the relative interval skill score (rISS). Δr, Nash-Sutcliffe efficiency-based assimilation index (NSEAI), and rISS are shown at the bottom of each panel.*

2. Line 40. I think it is better to change GHM definition only to Global Hydrodynamic Models instead of Hydrological due to some features the authors discuss further such as "runoff as a forcing factor", "discretized river", "surface water dynamics", etc.. Line 77 and 519 could be GHM instead of global hydrodynamic models.

Reply:

We would like to express our gratitude. We agree with referee #2 that our introduction should be more focused on global hydrodynamic models. We revised the sentence as follows.

*"As a result of recent computational advances, global hydrodynamic models (GHMs) have been used extensively to study the terrestrial water cycle (Döll et al., 2016; Sood and Smakhtin, 2015)."*

3. Line 52. The authors could also mention Laser altimetry. The ICESat missions are also very used in academic research. Maybe instead of a radar pulse, can be a radar/light pulse or even an electromagnetic pulse.

   Reply:

   We would like to thank referee #1 for suggesting introducing laser missions. So, we revised the text as below.

   *"Satellite altimetry quantifies the water surface elevation (WSE) by measuring the time required for the electromagnetic (e.g., radar or laser) pulse to travel between the satellite and the water surface."*

4. As an alternative to the semi-variogram analysis to determine the spatial dependency weights, the authors could have used "backwater lengths in rivers" studied by Samuel (1989). It can give an idea of which river reaches are affected by WSE variations at the VS locations. It would be a good idea to compare both approaches in future studies (not now).

   Samuels, P. G. (1989). Backwater lengths in rivers. Proceedings of the Institution of Civil Engineers, 87(4), 571–582. https://doi.org/10.1680/iicep.1989.3779

   Reply: We would like to express our gratitude to referee #2 for the suggestion. We will consider the suggestion in our future studies.

5. Line 156. using a power law dependent on what? Width? Upstream drainage area? Is it the same power law parameters for the whole basin?

   Reply: We appreciate referee #2 for raising this question. The power law depends on a prior annual average river discharge and uses a single set of parameters (i.e., *a* and *b*) for the whole basin (*a=0.1, b=0.5*). The power law was shown in Equation 1 in the supplementary document. We revised the text to reflect this comment as follows.

   *"The river channel depth was estimated using a power law relationship with prior river discharge (Supplementary Text S2, Equation 1) (Yamazaki et al., 2011; Zhou et al., 2022).*

6. Line 199. Something went wrong with the font size of some words. Line 351. Line 438. Line 637.

   Reply: We would like to thank referee #2 for identifying the mistakes. We corrected all those kinds of font errors.

7. Line 229. So, the mean and standard deviation were calculated based on the open loop simulation?

   Reply: Thanking referee #2, the mean and standard deviation were calculated based on the long-term open-loop simulation (2000-2014).

8. Line 231 to 239. Some of your readers might be unfamiliar with the Amazon Basin. It would be interesting to write a short and objective section about this basin, presenting a DEM map at least (a mean Precipitation map would be nice too).

Reply: We like to thank referee #2 for the great suggestion. We are happy to include a figure of the Amazon basin but considering the length of and the number of figures in the manuscript, we added them to the supplementary material.

[Figure]

*Figure S1: Amazon basin with elevations indicated by colors. River network is indicated by blue.*

We added some descriptions about the Amazon basin as follows.

*"The Amazon River basin is a major hydrological system containing a variety of rivers, floodplains, and wetlands (Reis et al., 2019). It contains four of the world's largest rivers, namely Solimões-Amazonas, Madeira, Negro, and Japurá rivers. The Amazon Basin receives a high annual rainfall of 2,200 mm/year where 30%-40% of the rainfall is recycled locally through evapotranspiration (Fassoni-Andrade et al., 2021). The Amazon River flows into the Atlantic Ocean with an average annual discharge of 206×10^3 m^3 s^(-1) accounting for about 20% of total world freshwater reaching the ocean yearly (Fassoni-Andrade et al., 2021). Amazon River serves a variety of human needs, including fluvial transportation, agriculture, fishing, and energy generation. The Amazon basin has seen substantial changes in hydrological severe events such as floods and droughts in recent years, with documented increases in amplitude and frequency due to increased rainfall intensity. Furthermore, there is an obvious pattern of an increasing frequency of severe floods in the northern and main stem regions and an upward trend of extreme drought events in the southern regions (Wongchuig et al., 2019)."*

9. Line 275. How do you measure the relative sharpness and the difference in reliability? Line 383 should be here.

Reply: Thank you very much for the suggestion. We included equations to calculate the relative sharpness and difference in reliability in the revised manuscript. Following are the revised text and added equations.

*"Furthermore, relative sharpness (rSharpness) and difference in reliability (ΔReliability) were used to evaluate relative assimilation performance. rSharpness is calculated as*

$$rSharpness = \frac{Sharpness_{asm} - Sharpness_{opn}}{Sharpness_{opn}} \qquad (1)$$

$$Sharpness = u - l \qquad (2)$$

*where Sharpness$_{asm}$ and Sharpness$_{opn}$ are Sharpness values of the assimilated and open-loop simulations, respectively. ΔReliability is defined as*

$$\Delta Reliability = Reliability_{asm} - Reliability_{opn} \qquad (3)$$

*where Reliability$_{asm}$ and Reliability$_{opn}$ are Reliability values of the assimilated and open-loop simulations, respectively. Reliability is the number of observations within the u and l bounds."*

10. Several wrong references in section 3.1.

    Line 309. The authors said that the Santos Dumont gauge is in the Purus River, but in Figure 4 it says Jurua River.

    Line 332. "Figure 5c–e displays hydrographs of the Jurua (Gaviao), Amazon (Manacapuru), and Negro (Serrinha) rivers" but in Figure 5 it says Manicore on the Madeira River, Aruma on the Purus River, and Sao Felipe on the Negro River.

    Line 351. "The lower panels of Figure 6 illustrate flow dynamics along the Amazon mainstem (Sao Paulo De Olivenca; Figure 6c) and Japura (Vila Bittencourt; Figure 6d) and Negro (Curicuriari; Figure 6e) rivers." but in Figure 6 it is written "Hydrographs recorded at Humaita on the Madeira River, Santos Dumont on the Jurua River, and Canutama on the Purus River are presented on panels c, d, and e, respectively."

    *Reply: We would be grateful to the referee for identifying the mistake. We revised Figures 4, 5, and 6 to correspond to the description of Sections 3.1.1., 3.1.2., and 3.1.3. The revised figures are included in the answer to comment #1.*

11. Line 321. Saying that the "direct DA generally improved flow dynamics" is very optimistic. Based on these results, I'd probably say that the direct DA maintained or even degraded the general performance, at least for discharge.

    *Reply: We would be thankful to the referee for the detailed comments on the text. We agree with the referee that the Direct DA method somewhat degraded the performance of discharge estimations in some locations. But the discharge estimates were improved in some gauge locations in the Amazon basin such as Santo Antonio Do Ica and Sao Paulo de Olivenca. When satellite altimetry observations are within comparable limits of the hydrodynamic simulation, the direct DA performed well. Therefore, we revised the sentence to deliver the meaning suggested by referee #2.*

    *"In summary, direct DA improved flow dynamics when the simulations were within comparable limits with satellite observations."*

12. Line 340. Once again, I think it is an optimistic conclusion. In the last sentence, the authors just said: "although NSE and ISS values worsened slightly." So how can the authors say afterward that "discharge estimates improved moderately"? I don't think that improvements in the correlation coefficient are enough for such a statement given that the NSE has become worst. But I reckon that seasonality got better as correlation got higher. Maybe the authors should clarify what they try to achieve with DA assimilation.

Reply: We would like to express our gratitude to referee #2. Our goal was to improve the overall performance of the river discharge hence higher NSE the better the discharge estimate would be. But we try to look at the positives of each method and come to a broader conclusion as none of the methods performed perfectly in all the scenarios. Ideally, it is better if the direct DA performs better than others as assimilation will not depend on prior statistics of the open loop simulation. We basically discuss the median performance in the discussion of NSE and ISS here. But there is a large variation in the statistics as shown in figure 7, table 2 and figure S2. A reasonable number of gauges improved their performance by anomaly DA. An overall moderate number of gauges (35%) improved their discharge estimations. Therefore, we modified the text to highlight the percentage of improved gauges.

*"Overall, the discharge estimates improved in some GRDC gauging stations (35%) with the assimilation of WSE anomalies into the hydrodynamic model."*

13. Figure S2 should be in the main manuscript. It could be together with Figure 7 as 7c, 7d, and 7e. Figures 7a and 7b don't need to be so large.

Reply: We would like to appreciate referee #2 for the valuable suggestion. We included the rISS panel of Figure S2 to Figure 7 considering the number of figures in the manuscript.

[Figure]

*Figure 7: a) Cumulative distribution of the correlation coefficient (Δr) for each experiment, shown in blue, yellow, and red for direct (DIR), anomaly (ANO), and normalized value (NOM) experiments, respectively. b) Boxplots of the Nash-Sutcliffe efficiency based assimilation index (NSEAI) and c) relative Interval Skill Score (rISS) of assimilated compared to open-loop discharge for all the experiments. Boxes in blue, yellow, and red indicates direct (DIR), anomaly (ANO), and normalized value (NOM), respectively.*

14. Line 427. "However, the direct DA experiments efficiently improved sharpness, thereby increasing confidence in the assimilated river discharge." I would say "FALSELY

increasing confidence" as the authors just observed that the reliability drops more than 50% for direct DA experiments. What is the point of being narrower if the observation falls out of the confidence interval? I think the authors should be careful with that.

Reply: We would like to thank referee #2 for pointing out this. We agree with the referee that "falsely" increasing confidence will not be beneficial. In this sentence, we meant that when data assimilation is performed in direct values, the spread of final assimilated values will be narrower than in other methods because the assimilation was performed in anomalies or normalized values. Of course, the reliability should be higher. We revised the sentence to convey the idea more clearly as follows.

*"However, the direct DA experiments efficiently improved sharpness, thereby increasing confidence in the assimilated river discharge when reliability is higher."*

15. On tables 3 and 4 it would be nice to see the Open Loop and the CaMa VIC BC performances for comparison.

Reply: Thank you very much for the nice suggestion. We included the median statistics of the Open Loop and the CaMa VIC BC in tables 3 and 4 even though CaMa VIC BC cannot be assessed for sharpness. We include the revised tables below.

*Table 3: Median performance metrics for daily discharge estimates obtained from DA experiments. Median values for the correlation coefficient (r), Nash-Sutcliffe efficiency (NSE), Kling-Gupta efficiency (KGE), and width of the confidence interval (Sharpness) are presented for all GRDC gauges and gauges in the satellite coverage area.*

| Experiment | All | | | | Satellite Converge Reaches | | | |
|---|---|---|---|---|---|---|---|---|
| | $r$ | NSE | KGE | Sharpness $(10^6)$ | $r$ | NSE | KGE | Sharpness $(10^6)$ |
| Open-Loop | 0.83 | 0.50 | 0.59 | 1.17 | 0.91 | 0.71 | 0.77 | 1.59 |
| DIR | 0.74 | 0.13 | 0.46 | 1.09 | 0.88 | 0.21 | 0.48 | 5.79 |
| ANO | 0.85 | 0.39 | 0.55 | 1.18 | 0.95 | 0.66 | 0.70 | 13.91 |
| NOM | 0.84 | 0.50 | 0.62 | 1.17 | 0.95 | 0.76 | 0.72 | 14.37 |
| CaMa VIC BC | 0.81 | 0.42 | 0.60 | - | 0.91 | 0.68 | 0.76 | - |

*Table 4: Median performance metrics for water surface elevation estimates obtained from DA experiments. Median values for the root mean square error (RMSE), long-term bias (BIAS), and difference in amplitude (ΔA) are presented for all, assimilation, and validation VSs.*

| Experiment | All | | | Assimilation | | | Validation | | |
|---|---|---|---|---|---|---|---|---|---|
| | RMSE | BIAS | ΔA | RMSE | BIAS | ΔA | RMSE | BIAS | ΔA |
| Open-Loop | 4.60 | 2.43 | 1.98 | 4.65 | 2.49 | 1.92 | 4.45 | 1.98 | 2.29 |
| DIR | 4.56 | 2.38 | 3.86 | 4.58 | 2.39 | 3.68 | 4.25 | 1.90 | 4.75 |
| ANO | 4.80 | 2.82 | 1.60 | 4.79 | 2.83 | 1.53 | 4.88 | 2.69 | 2.08 |
| NOM | 4.80 | 2.76 | 1.75 | 4.78 | 2.74 | 1.72 | 4.96 | 2.84 | 2.10 |
| CaMa VIC BC | 4.89 | 3.14 | 2.51 | 4.38 | 3.02 | 2.37 | 5.12 | 3.35 | 3.46 |

16. Line 541. HTESSEL not HTEESSEL.

Reply: Thank you for recognizing the mistake. We modified HTEESSEL to HTESSEL as shown below.

*"For simplicity, we used only a single runoff (HTESSEL; Balsamo et al., 2011) from E2O WRR2 to prepare the runoff ensemble."*

17. Line 624. Which experiment is that? The one in section 4.2.? Or the one with VIC BC (section 3.3)?

Reply: Thank you for recognizing the mistake. It should be VIC BC. We corrected the sentence as shown below.

*"River discharge was well characterized in the normalized value assimilation experiments, with a median NSE≈0.47, which was better than the river discharge produced by the uncalibrated model with default parameters using VIC BC runoff (Lin et al., 2019) (median NSE≈0.13)."*

Referee #3

This manuscript explores different strategies to assimilate water surface elevation (WSE) derived from satellite altimetry into the CaMa-Flood global hydrodynamic model. During the recent years, a large number of studies have demonstrated the potential benefits of assimilating WSE for various purposes, such as parameter estimation or river dynamics modeling improvement. One of the main difficulties relies on the fact that WSE from altimetry provides the water elevation based on a reference (geoid) that can differ from the model reference (because of DEM errors for example) and the induced bias may highly degrade the assimilation performances. An alternative consists in considering WSE anomalies instead of absolute values. Another one, introduced in this study, also normalizes WSE anomalies to account for possible errors in the signal amplitude. Here the authors explore those three strategies and their respective performances in reproducing river discharge over the Amazon Basin. This scientific question is highly relevant and the study is well conducted, which makes it worth publishing, especially in the context of the forthcoming SWOT mission.

Overall, the manuscript is well written and well organized. The methodology is clearly stated and the results are quite convincing, although the latter should be revisited to correct some mistakes and clarify some points, as explained thereafter.

Reply:

We would like to convey our acknowledgment to referee #3 for his informative comments. We will address all the comments in the revised text, and responses are provided below.

Major remarks

1.  The quality of figures 4 to 6 is quite bad and colors are hard to identify (in the maps and in the time series). Also the time series subplots do not correspond to the description in the results section (3.1.1, 3.1.2 and 3.1.3). Hence it is not possible understand, confirm or refute the description of these plots, as well as the conclusions drawn (L309-323, L332-342, L351-362).

    Reply:

    We would be grateful to referee #3 for raising this issue. When creating these figures, a mistake occurred. Therefore, we revised figures 4-6 to correspond to the descriptions in sections 3.1.1, 3.1.2, and 3.1.3.

[Figure]

*Figure 4: a) Difference in the correlation coefficient of river discharge (Δr) and b) relative Root Mean Square Error (rRMSE) of water surface elevation for DIR. Circles indicate virtual stations used for data assimilation, and squares are virtual stations used for validation on the WSE plots. Hydrographs recorded at Labera on the Purus River, Santos Dumont on the Jurua River, and Santo Antonio Do Ica on the Amazon River are presented in panels c, d, and e, respectively. The locations of the hydrographs shown in panels c, d, and e are presented in panel a. Discharge observations are shown in black, assimilated simulation results in orange, and open-loop simulation results in blue. The color range indicates the 95% confidence interval used to calculate the relative interval skill score (rISS). Δr, Nash-Sutcliffe efficiency-based assimilation index (NSEAI), and rISS are shown at the bottom of each panel.*

[Figure]

*Figure 5: a) Difference in the correlation coefficient of river discharge (Δr) and b) relative Root Mean Square Error (rRMSE) of water surface elevation for DIR. Circles indicate virtual stations used for data assimilation, and squares are virtual stations used for validation on the WSE plots. Hydrographs recorded at Gaviao on the Jurua River, Manacapuru on the Amazon River, and Serrinha on the Negro River are presented on panels c, d, and e, respectively. The locations of the hydrographs shown in panels c, d, and e are presented in panel a. Discharge observations are shown in black, assimilated simulation results in orange, and open-loop simulation results in blue. The color range indicates the 95% confidence interval used to calculate the relative interval skill score (rISS). Δr, Nash-Sutcliffe efficiency-based assimilation index (NSEAI), and rISS are shown at the bottom of each panel.*

[Figure]

*Figure 6: a) Difference in the correlation coefficient of river discharge (Δr) and b) relative Root Mean Square Error (rRMSE) of water surface elevation for DIR. Circles indicate virtual stations used for data assimilation, and squares are virtual stations used for validation on the WSE plots. Hydrographs recorded at Sao Paulo De Olivenca on the Amazon River, Vila Bittencourt on the Japura River, and Curicuriari on the Negro River is presented on panels c, d, and e, respectively. The locations of the hydrographs shown in panels c, d, and e are presented in panel a. Discharge observations are shown in black, assimilated simulation results in orange, and open-loop simulation results in blue. The color range indicates the 95% confidence interval used to calculate the relative interval skill score (rISS). Δr, Nash-Sutcliffe efficiency-based assimilation index (NSEAI), and rISS are shown at the bottom of each panel.*

2. It is not clear to me which variables are included in the state vector x. I understand from L509 that the prognostic variable of CaMa-Flood is water storage. On the other hand, it is stated (L188) that the state vector includes river discharge, WSE, flooded area, flood height and storage. Could you clarify this point? Also, in the latter case, shouldn't the observation operator H contain only zeros except for the column corresponding to WSE? Moreover, are all state variables (river discharge, WSE, etc.) converted to anomaly and normalized values as written in L193?

Reply:

We would like to express our gratitude to referee #3 for the insightful comment. We think that using the same H (observational operator) in both equations 2 and 3 is confusing. Equation 1 is about the CaMa-Flood model time evaluation and equation 2 is the conceptual relationship of CaMa-Flood state variables with the observations. So, the x (simple x) vector consists of CaMa-Flood state variables. Equation 3 is the analysis equation for data

assimilation, there we used WSE only for data assimilation. Therefore, we think H in equation 3 is a subset of H in equation 2. Hence, we revised the H in equation 2 to be $\mathcal{H}$ (curly H) and revised the text to explain that we used only WSE in the LETKF analysis equation. The following text presents the changes made to the revised manuscript.

*"The LETKF is a commonly used DA algorithm (e.g., Feng et al., 2021; Ishitsuka et al., 2020; Revel et al., 2019, 2021b; Wongchuig-Correa et al., 2020) for nonlinear models, which are needed for modeling hydrodynamic processes. The nonlinear hydrodynamic model can be shown in discrete form as follows:*

$$x_{k+1} = \mathcal{M}(x_k, u_k, \vartheta) + q_k, \tag{4}$$

*where $x$, $u$, and $\vartheta$ represent the vector of the state variable, model forcing, and model parameters, respectively. The nonlinear model operator, $\mathcal{M}$, is related to the time interval of $t_k$ to $t_{k+1}$, whereas errors in the model structure, parameters, forcing, and antecedent states are represented by $q_k$. All state variables in CaMa-Flood, such as river discharge, WSE, flooded area, flood height, and storage, are included within the vector $x$. The model states can be related to the observations as follows:*

$$y_k = \mathcal{H}(x_k) + \varepsilon_k, \tag{5}$$

*where $y$ is the observation vector; $\varepsilon$ is the vector of observation errors; and $\mathcal{H}$ is the linear observation operator, which relates the model states ($x$) to the observations ($y$). In this study, the observations were WSE obtained from satellite altimetry. In the anomaly and normalized value assimilations, the observed and forecasted states were transformed into anomalies and normalized values, respectively (Section **Error! Reference source not found.**, **Error! Reference source not found.**). The LETKF assimilation algorithm was used to obtain the optimal estimate of the model state variable X considering the model and observation errors. Here, the model state variable X was composed of WSE. LETKF analysis is expressed as*

$$X^a = \overline{X^f} + E^f \left[ VD^{-1}V^T (HE^f)^T \left(\frac{R}{w}\right)^{-1} (Y^o - \overline{HX^f}) + \sqrt{m-1} VD^{-1/2}V^T \right], \tag{6}$$

*where $X^a$ is the posterior state estimator (or analysis), $X^f$ is the prior state estimator (or forecast), $Y^o$ is the observation (i.e., the WSE value obtained from satellite altimetry), H is the observation operator correspond to WSE which is a subset of $\mathcal{H}$, $m$ is the ensemble size, $E^f$ is the prior state error covariance obtained directly from the perturbations, R is the observation error covariance determined from the uncertainty of the measurements, $w$ is the weighting term for observation localization calculated with semi-variogram analysis of the simulated WSE (Revel et al., 2019), and $VDV^T$ is defined as*

$$VDV^T = (m-1)I + (HE^f)^T R^{-1} HE^f \tag{7}$$

*where I is the unit matrix of dimension $m \times m$, representing the number of perturbations. $VD^{-1}V^T$ and $VD^{-1/2}V^T$ are calculated through eigenvalue decomposition of $VDV^T$. The overbar represent the ensemble mean vector."*

3. More importantly, I think that the analysis of DA performances when runoff forcing or bathymetry are biased (section 4.2) requires a bit more explanations.

Reply:

We would like to express our gratitude to referee #3 for the valuable suggestions regarding Section 4.2. We will answer each comment one by one. We found that we have made some errors in making figure 11. Hence, we revised figure 11 as shown below.

[Figure]

*Figure 11: Comparison of Nash-Sutcliffe efficiency (NSE) of assimilated river discharge under various conditions: a) without runoff bias or bathymetry error, b) without runoff bias and with bathymetry error, c) with runoff bias and without bathymetry error, and d) with runoff bias and bathymetry error. The direct, anomaly, and normalized value DA results are represented in blue, yellow, and red, respectively.*

- Model simulations are affected by biases (errors in absolute values) and by errors in dynamics. Decreasing the river bathymetry (by lowering the river bottom elevation) would lower the absolute WSE without impacting the flow dynamics, except if bank overflow occurs (flooding). If there is no flooding, I would have expected large impacts on the direct DA performances but no impact on anomaly and normalized value DA. Given that, do the results of Fig. 11b mean that the degradation of those two experiments is due to bank overflow? In addition, what can explain the very poor performances of normalized value DA method? Maybe some example time series could help better understand these results, as done for the previous experiments (perhaps as a supplement).

Reply:

We would like to thank referee #3 for the valuable comment and suggestion. In these experiments, we try to assess the performance of DA methods in simplified error conditions in forcing and parameters of the model. We agree with referee #3 that including corrupted bathymetry will lower the absolute WSE. Hence, it has a visible effect on WSE assimilation in direct DA.

But in the cases of anomaly and normalized value DA, the degradation of discharge accuracy is mainly due to the errors in the statistics (i.e., mean and standard deviation) which were used to convert the absolute WSE values to anomalies and normalized values. These statistics were computed using long-term open-loop simulations. When the river bathymetry is corrupted the mean open-loop WSE will also be biased, and the standard deviation will be different. Hence the assimilation result will also be inaccurate.

- Is there any possible explanation of the better normalized value DA performances with runoff bias and bathymetry error compared to performances with only bathymetry errors (panels b and d)?

Reply:

We would like to express our gratitude to referee #3 for raising this question. With current experiments, the statistics of the normalized value DA method should be much lower

[Figure]

*Figure S7: Hydrographs of assimilated river discharge under various conditions: a) without runoff bias or bathymetry error, b) without runoff bias and with bathymetry error, c) with runoff bias and without bathymetry error, and d) with runoff bias and bathymetry error. The direct, anomaly, and normalized value DA results are represented in blue, yellow, and red, respectively.*

because of the lowering of bathymetry as well as reducing the runoff. We have checked the experimental setting again and found that there was an error in preparing Figure 11. The revised figure shows low *NSE* values for anomaly and normalized value DA in the experiment setting with errors in runoff and corrupted bathymetry. The underestimation of river discharge is evident by comparing the representative hydrograph added to the revised supplementary material.

- Finally, considering only one runoff (HTESSEL) to generate the ensemble reduces the dynamics variations between the members. Could this be a reason of the poor DA performances, especially with the normalized value DA method?

Reply:

We would like to express our sincere thanks to referee #3. Even though we agree with referee #3 that using one runoff reduces the perturbation mixing, we do not think the poor performance of the normalized value DA method in corrupted bathymetry or runoff error cases is due to low variation in perturbations. It was basically due to the bias and differences in statistics used for anomaly and normalized DA method. Therefore, the biggest limitation of anomaly or normalized DA methods was that those are dependent on the statistics (i.e., mean and standard deviation of open-loop simulation) used to generate anomalies or normalized values.

The modified figure 11 represents a more logical representation of the error conditions of the hydrodynamic modeling. Therefore, we revised the text as shown below.

*"Runoff bias and bathymetry errors affect the accuracy of assimilated river discharge in different ways and Figure 11 compares boxplots of NSE values for different DA methods with different error conditions. When neither runoff nor bathymetry was erroneous, the normalized value DA method performed best (median NSE=0.83) at estimating river discharge in terms of NSE (Figure 11a). When the bathymetry contained some errors, but runoff was unbiased (Figure 11b) none of the DA methods were able to improve the river discharge from open-loop simulation. The performance of river discharge was not much affected in the open-loop simulation by the river bathymetry corruption as river discharge is not much influenced by the river bathymetry (Modi et al., 2022). But when the satellite altimetry was assimilated into the hydrodynamic model the accuracy of the river discharge estimates was degraded. In the direct DA, assimilated WSE values were higher than the observed WSE resulting in river discharge overestimation when the bathymetry has errors (Supplementary Text S5, Figure S7). In contrast, anomaly and normalized value DA was affected by the bias in the open-loop statistics (i.e., mean and standard deviation) used for generating anomalies and normalized values (Supplementary Text S5, Figure S7). Bias in the runoff ensemble strongly affected the accuracy of river discharge estimation with anomaly and normalized value DA, as bias in runoff causes bias in the mean and standard deviation used to generate WSE anomalies and normalized values (Figure 11c). Direct DA provided the best discharge estimation (median NSE=0.68) when runoff was biased. When both runoff and river bathymetry were erroneous, none of the DA methods produced better discharge estimates than open-loop simulation. Therefore, the normalized DA method worked well under normal conditions, but anomaly DA produced better discharge estimates when the river bathymetry had errors, and the direct DA method performed best under runoff-biased conditions. Simple calibration of the hydrodynamic model is*

*recommended for successful normalized value DA (i.e., bias correction of runoff to obtain the mean discharge and river bathymetry calibration to accurately determine the mean WSE)."*

In addition, we added some descriptions to the supplementary materials as *"Text S5: Data assimilation under various conditions"* as shown below.

*"We compared the DA performances with different hydrodynamic model conditions and combinations of them. We tested biased runoff condition ("with runoff bias": Text S2); corrupted bathymetry condition ("with bathymetry error": Text S3); and a combination of biased runoff and corrupted bathymetry. Figure S7 indicates the hydrograph of Mancapuru gauging station for different DA methods and different model conditions. The normalized assimilation method estimated the discharge closer to the observed discharge neither with any runoff bias nor bathymetry error. But when the bathymetry error is imposed none of the assimilation methods performed better than the open-loop simulation. When the runoff was erroneous the direct DA method performed better in estimating river discharge. The main reason for poor performance in anomaly and normalized value DA methods especially in bathymetry error conditions is the underestimation of the open-loop statistics (i.e., mean and standard deviation). The better performance of direct DA with runoff bias and without bathymetry error can be due to no error in river channel parameters (i.e., river bathymetry). Therefore, the poor performance of the anomaly and normalized value DA methods is due to the errors in the statistics used to generate anomalies and normalized values in the erroneous hydrodynamic model."*

4. Each of the three DA methods can outperform the other two depending on the configuration, making the choice of the DA method quite difficult for further studies. I think providing more insight in these experiments might help readers better understand the pros and cons of each method.

Reply:

We would like to appreciate referee #3 for the great suggestion. As we did extensive testing on normalized DA in this study, we provide some recommendations on the usage of normalized DA. The normalized DA method should be used with an unbiased runoff and somewhat calibrated to capture the WSE pattern. Hence, we revised the text as below.

*"The estimation of river discharge using DA methods is variable and depends on the state of the runoff data (i.e., biased runoff state) and the accuracy of river cross-sectional parameters (e.g., river bathymetry). In the current condition of the hydrodynamic model with perturbed HTESSEL runoff from the E2O WRR2 dataset, the normalized value DA method performed best among other DA methods. But when the runoff was biased without river bathymetry error, the direct DA approach performed best. However, when the river bathymetry was erroneous, none of the DA methods performed better than open-loop simulation. Hence, different DA approaches should be used depending on the runoff and river bathymetry. To realize the advantages of the normalized value DA approach, basic model calibration is necessary, such as calibration of runoff to capture the mean discharge and moderate calibration of bathymetry to capture WSE patterns."*

Minor remarks

1. Fig. 1. In panel b, upper left square, it should be "Altimetry Auxillary Data".

   Reply: Thank you for your recognition of the error. We corrected the text accordingly. The revised figure 1 is as follows.

[Figure]

   *Figure 1: a) Data assimilation framework, b) schematic diagram of satellite altimetry preprocessing, and c) derivation of the localization parameters.*

2. L134-137. "VSs with considerable variation in mean WSE compared to the MERIT Hydro (Yamazaki et al., 2017, 2019) elevation (expressed as riverbank height) were filtered through comparison of mean observations and riverbank heights." What could be the cause(s) of such errors? Maybe the answer is given in L506-508.

   Reply: We would like to thank referee #3 for raising this question. From our analysis, we found that most erroneous VSs are in narrower rivers at high elevations. There can be several reasons for these, 1. Non-nadir direction observations, 2. Errors in post-processing of VS (e.g., geoid conversion), etc. We were doing some further analysis on the quality of the satellite altimetry data. The VSs explained in L134-137 are filtered out and not used in the assimilation. The VSs which can be relatively compatible with the hydrodynamic model output was used for assimilation.

3. L157. Is the river width from remote sensing available for every reaches of the river network? If not, how is it determined?

   Reply: We thank referee #3 for the question. Remote sensing river widths were used in the river reaches with river width > 300m (Yamazaki et al., 2014). A power law relationship of the average river discharge was used to estimate the river width for the other smaller river reaches. We revised the text to reflect these ideas as follows.

   *"River widths were determined using remote sensing for rivers wider than 300m (Yamazaki et al., 2014a) and for narrower rivers river width was empirically determined (Yamazaki et al., 2011)."*

4. Fig. 2. I would suggest to add a legend in the time series, and maybe add error bars in observed WSE (from HydroWeb).

   Reply: Thank you very much for the suggestion. We revised Fig 2 according to referee #3's suggestion.

5. Eq. (2). Since H is linear, maybe it is better to write Hxk instead of H(xk).

   Reply: Thank you for the suggestion. Here H is more of an operator for converting simulated variables to observable variables. Not all the simulated variables are possible to observe. All the variables may not have a linear relationship to observation as WSE. So, we would like to keep equation (2) as it is, but we will use two different symbols for equations (2) and (3) because H in equation (3) is a subset of H in equation (2). The revised equations are shown in the answer for major remarks #2.

6. L288. Sharpness and reliability are not defined.

   Reply: Thank you for the suggestion. We defined sharpness and reliability in the methodology section. The text was revised as follows.

   *"Furthermore, relative sharpness ($rSharpness$) and difference in reliability ($\Delta Reliability$) were used to evaluate relative assimilation performance. $rSharpness$ is calculated as*

   $$rSharpness = \frac{Sharpness_{asm} - Sharpness_{opn}}{Sharpness_{opn}} \qquad (8)$$

   $$Sharpness = u - l \qquad (9)$$

   *where $Sharpness_{asm}$ and $Sharpness_{opn}$ are Sharpness values of the assimilated and open-loop simulations, respectively. $\Delta Reliability$ is defined as*

   $$\Delta Reliability = Reliability_{asm} - Reliability_{opn} \qquad (10)$$

   *where $Reliability_{asm}$ and $Reliability_{opn}$ are Reliability values of the assimilated and open-loop simulations, respectively. Reliability is the number of observations within the u and l bounds."*

7. L285. Nash and Sutcliffe (1970) and Kling and Gupta (2009) are cited several times, which is not necessary.

   Reply: Thank you. We removed the extra citations.

8. L313. It is written (but I cannot verify it) that the 95 % ensemble spread is improved until mid-2010, when the ENVISAT satellite was available. But this satellite is supposed to be available until 2012 (Tab. 1). Also, could you explain what an improvement in ensemble spread is? Is it a reduction of the spread?

   Reply: We appreciate referee #3 for the thorough reading of the manuscript. Even though ENVISAT was operational until 2012, the nominal period is 2002-2010 after the 2010 ENVISAT track was changed. Therefore, ENVISAT data after 2010 was not used in

HydroWeb as the VS locations were changed. Hence, we revised Table 1, we added only the nominal periods used in HydroWeb for each satellite.

The second part of the comment is about the definition of *"improvement in the ensemble spread"*. An improvement in ensemble spread is referred to as the reduction of spread or reduction of sharpness. We think *"improvement in ensemble spread"* is better to be revised as *"an improvement in sharpness"*. Hence, we revised the manuscript according to referee #3's comments as shown below.

9. L321. Considering numbers in L300-302, I would not say that "direct DA generally improved flow dynamic simulation to a moderate extent": concerning river discharge, 8 % of gauges show an improvement while 43 % show a degradation.

   Reply: We would like to express our gratitude to referee #3 for pointing out this. It was evident that many gauges degraded their accuracy of river discharge by the direct DA method. However, some gauges such as Santo Antonio Do Ica and Sao Paulo de Olivenca improved their discharge accuracy. Hence, we revised the text according to the comment of referee #3 to reflect that the river discharge estimated of some gauges shows improvement.

10. L326. What could be the impact of choosing different time periods for observed and simulated WSE when computing the long term mean (and std)? For example if a multi-year drought is accounted for in one period and not in the other.

    Reply: We would like to thank referee #3 for the great question. We believe that there cannot be much effect of using different time periods for statistics for simulation and observations as long as the statistics are representative statistics of the long-term values. In the current study, we used in fact different time periods for calculating statistics for simulations (e.g., 2009-2014) and observations (i.e., available period). But if the statistics are largely different from the observed statistics as shown in the biased runoff experiment and corrupted bathymetry, the accuracy of estimated discharge by the anomaly and normalized value DA can be hampered. We checked sensitivity on the simulation WSE statistics where we found that the statistics calculated using simulation of 5 years or more are reasonably reproduced river discharges.

11. L336. It should be "improvements in r and ISS" not "in Dr and rISS".

    Reply: Thanking referee #3, we revised the text as below.

    *"River discharge in the Amazon mainstem, notably at Manacapuru gauge (Figure 5d), was well characterized, with improvements in r and ISS but deterioration of NSE values."*

12. L348. Considering the quality of the figure and the color range, decreases in discharge correlation is not that evident in the Amazon mainstem.

    Reply: We would like to thank referee #3 for the comment. We will improve the quality of the figures in the revised manuscript. We agreed with referee #3 that the correlation in the Amazon mainstream is quite good. This may be because model parameters (e.g., bank full height, river bathymetry, etc.) are still very good on these river reaches.

13. L365. It is stated that the assimilation has very little influence outside the area of satellite observations. First, does the satellite coverage area correspond to the reaches downstream any VS? Second, shouldn't the localization method used here allow to correct river discharge upstream VS?

Reply: We would like to express our gratitude to referee #3 for the great question. Here, we define the satellite coverage area as the river reaches located downstream of the most upstream VS in each tributary (green circles in Figure S1 indicate GRDC locations that were in the satellite coverage area). Hence, the satellite coverage areas cover downstream of all the VSs. Although using the localization method, the WSE can be updated without using any local observation. But in the upper reaches (narrow rivers with small catchment areas), the size of the local patches is small (Revel et al., 2019). Hence, assimilation efficiency can be lower in these upper reaches compared to downstream rivers.

14. Fig. 7. In the caption of a, it should be "probability distribution" instead of "cumulative distribution". Also, it could be helpful to plot the vertical line at 0. Same remark for Fig. S5.

Reply: We would like to thank referee #3 for the suggestion. We revised the caption of Fig. 7 and Fig. S5 according to referee #3's suggestion.

For Fig. 7:
*"Figure 7: a) Probability distribution of the correlation coefficient (Δr) for each experiment, shown in blue, yellow, and red for direct (DIR), anomaly (ANO), and normalized value (NOM) experiments, respectively. b) Boxplots of the Nash-Sutcliffe efficiency based assimilation index (NSEAI) and c) relative Interval Skill Score (rISS) of assimilated compared to open-loop discharge for all the experiments. Boxes in blue, yellow, and red indicates direct (DIR), anomaly (ANO), and normalized value (NOM), respectively."*

For Fig. S5:
*"Figure S5: a) Probability distribution of the correlation coefficient (Δr) for each experiment, shown in blue, yellow, and red for direct (Exp 1a), anomaly (Exp 2a), and normalized value (Exp 3a) using calibrated model, respectively. b) Boxplots of the Nash-Sutcliffe based assimilation index (NSEAI) of assimilated compared to open-loop discharge for all the experiments. Boxes in blue, yellow, and red indicates direct (Exp 1a), anomaly (Exp 2a), and normalized value (Exp 3a) using calibrated model, respectively."*

15. L385. "A large reduction in sharpness was observed in the direct assimilation experiment (Exp 1), mainly because the assimilation was conducted directly." I do not see the link here, could you expand a bit more?

Reply: Thank you very much for asking for calcification. Sharpness reduction means the reduction of the ensemble spread. When the assimilation was performed in real values the ensemble spread become smaller than it was performed in anomaly or normalized values. In the direct DA method, the assimilated values were directly used to update the initial conditions of the next time step in the mode, but in anomaly and normalized DA methods, the assimilated values were reprojected to the WSE values using statistics which kind of redistribute the perturbations. Hence, direct DA potentially has a lower ensemble spread. We revised the text to reflect this meaning in the revised manuscript as follows.

*"A large reduction in sharpness was observed in the direct assimilation experiment (DIR), mainly because the assimilation was conducted using direct values in DIR (Figure 4c–e) whereas the ANO and NOM reprojected the assimilated values to WSE values."*

16. L408. For river discharge, sharpness is also considered (L415).

Reply: Thank you very much. We indeed calculated sharpness for river discharge as well even though we did not include it in Fig 8. The median sharpness values were shown in Table 3.

17. L530. Indeed, a huge potential from SWOT is expected in this kind of study. But how to deal with the need to compute long term mean and std for the derivation of anomalies and normalized values?

Reply: We would like to thank referee #3 for the valuable question. In this study, we did not assess the possibilities of using anomaly or normalized DA methods for the real-time forecast or usage with short-term observation records. However, a representative estimate for SWOT observations can provide reasonable discharge estimates. On the other hand, as SWOT data is 2-dimensional data unlike the satellite altimetry from other satellites, the observation closer to the river simulation pixel can be used. Some more discussion on the matter was provided in our previous paper (Revel et al., 2021). There is still some room for improvement in the current methodology for use to assimilate SWOT data in near real-time. We will assess the sensitivity of the observational statistics which can be used for the real-time/short-term forecast in our future studies.

18. L536. Water height in the river is approximately 50 % lower, not WSE.

Reply: Thank you very much for pointing out this. We revised the text accordingly as shown below.

*"Because of the bias introduced by the runoff forcing, river discharge was approximately 50% lower in the open-loop simulation than in the observations."*

19. Fig. 12. The range of the y-axis could be reduced.

Reply: Thank you very much. We revised Figure 12 by reducing the y-axis.

[Figure]

*Figure 12: Boxplot comparison of Nash-Sutcliffe efficiency-based assimilation index (NSEAI) values for uncalibrated and calibrated models with a) direct DA, b) anomaly DA, and c) normalized value DA.*

Minor remarks in supplementary material

1. Fig. S1. Square and circle are inverted in the legend.

   Reply: Thank you very much. We corrected it. In the revised supplementary Figure S1 became Figure S2.

[Figure]

● Inside Satellite Observations
■ Outside Satellite Observations

*Figure S2: GRDC gauges within Satellite coverage of ENVISAT and Jason1/2 in green circles where others indicated by red squares.*

2. L45. It should be "Exp 2a and Exp3a".

   Reply: Thank you for pointing this out. We revised it by using different naming convention as below.

   *"We denote the direct, anomaly, and normalized value DA experiments as DIR_2, ANO_2, and NOM_2, respectively."*

3. L57. What is Exp 3b?

   Reply: We would like to thank referee #3. Exp 3b should be removed. We revised the text as follows.

   *"We investigated further into the peak and low flow values of the normalized value assimilation experiments namely normalized value assimilation with normal conditions (NOM) and normalized assimilation to calibrated model (NOM_2); since they are important parameters of a hydrograph that directly affects floods and drought occurrences."*

4. Fig. S7. It is hard to see the effect of DA on low flows. Maybe consider a log-log scatter plot?

Reply: Thank you very much for the comment. We revised Figure S9 (in the revises supplementary Figure 7 became Figure 9), as shown below.

[Figure]

*Figure S9: Scatter plot of the simulated annual maximum and minimum compared to observed maximum river discharge for NOM and NOM_2. Circles and squares represents assimilated and open-loop river discharge, respectively.*

**Reference:**
Feng, D., Gleason, C. J., Lin, P., Yang, X., Pan, M., & Ishitsuka, Y. (2021). Recent changes to Arctic river discharge. *Nature Communications*, *12*(1), 1–9. https://doi.org/10.1038/s41467-021-27228-1

Ishitsuka, Y., Gleason, C. J., Hagemann, M. W., Beighley, E., Allen, G. H., Feng, D., et al. (2020). Combining optical remote sensing, McFLI discharge estimation, global hydrologic modelling, and data assimilation to improve daily discharge estimates across an entire large watershed. *Water Resources Research*, 1–20. https://doi.org/10.1029/2020wr027794

Lin, P., Pan, M., Allen, G. H., Frasson, R. P., Zeng, Z., Yamazaki, D., & Wood, E. F. (2020). Global Estimates of Reach-Level Bankfull River Width Leveraging Big Data Geospatial Analysis. *Geophysical Research Letters*, *47*(7), 1–12. https://doi.org/10.1029/2019GL086405

Revel, Ikeshima, Yamazaki, & Kanae. (2019). A Physically Based Empirical Localization Method for Assimilating Synthetic SWOT Observations of a Continental-Scale River: A Case Study in the Congo Basin. *Water*, *11*(4), 829. https://doi.org/10.3390/w11040829

Revel, M., Ikeshima, D., Yamazaki, D., & Kanae, S. (2021). A Framework for Estimating Global-Scale River Discharge by Assimilating Satellite Altimetry. *Water Resources Research*, *57*(1), 1–34. https://doi.org/10.1029/2020WR027876

Wongchuig-Correa, S., Cauduro Dias de Paiva, R., Biancamaria, S., & Collischonn, W. (2020). Assimilation of future SWOT-based river elevations, surface extent observations and discharge estimations into uncertain global hydrological models. *Journal of Hydrology*, *590*(March), 125473. https://doi.org/10.1016/j.jhydrol.2020.125473

Yamazaki, D., Ikeshima, D., Tawatari, R., Yamaguchi, T., O'Loughlin, F., Neal, J. C., et al. (2017). A high-accuracy map of global terrain elevations. *Geophysical Research Letters*, *44*(11), 5844–5853. https://doi.org/10.1002/2017GL072874